# A GREB1-steroid receptor feedforward mechanism governs differential GREB1 action in endometrial function and endometriosis

Sangappa B. Chadchan[1], Pooja Popli[1], Zian Liao[1], Eryk Andreas[2], Michelle Dias[3], Tianyuan Wang[4], Stephanie J. Gunderson[2], Patricia T. Jimenez[2], Denise G. Lanza [5], Rainer B. Lanz[6], Charles E. Foulds [7], Diana Monsivais[1], Francesco J. DeMayo [8], Hari Krishna Yalamanchili[3,9,10], Emily S. Jungheim[2,11], Jason D. Heaney [5], John P. Lydon[6], Kelle H. Moley[2], Bert W. O'Malley [6] & Ramakrishna Kommagani [1,12] ✉

Cellular responses to the steroid hormones, estrogen (E2), and progesterone (P4) are governed by their cognate receptor's transcriptional output. However, the feed-forward mechanisms that shape cell-type-specific transcriptional fulcrums for steroid receptors are unidentified. Herein, we found that a common feed-forward mechanism between GREB1 and steroid receptors regulates the differential effect of GREB1 on steroid hormones in a physiological or pathological context. In physiological (receptive) endometrium, GREB1 controls P4-responses in uterine stroma, affecting endometrial receptivity and decidualization, while not affecting E2-mediated epithelial proliferation. Of mechanism, progesterone-induced GREB1 physically interacts with the progesterone receptor, acting as a cofactor in a positive feedback mechanism to regulate P4-responsive genes. Conversely, in endometrial pathology (endometriosis), E2-induced GREB1 modulates E2-dependent gene expression to promote the growth of endometriotic lesions in mice. This differential action of GREB1 exerted by a common feed-forward mechanism with steroid receptors advances our understanding of mechanisms that underlie cell- and tissue-specific steroid hormone actions.

The nuclear receptor super-family is comprised of ligand-activated transcription factors that mediate distinct and complex physiological functions of numerous hormones, including but not limited to steroid, retinoid, and thyroid hormones[1–8]. The continuous advancement in our understanding of these proteins revealed the molecular cues that underpin the remarkable roles of steroid hormones in physiology and pathophysiologies[3,4]. Both ligand-activated as well as unliganded nuclear receptors bind to their cognate-responsive elements on promoters and recruit coregulators (coactivators and corepressors) to modulate the transcriptional output of target genes[1,3,4]. Importantly, the distinct expression of coactivators and effector proteins direct these tissue and cell-type specific actions of steroid hormones and their receptors[1,3,4]. Recent findings revealed additional mechanisms including tissue-specific target genes of hormones that mediate the receptor-driven actions[2–5]. However, such feed-forward mechanisms are not well explored for steroid hormones, specifically estrogen and

progesterone, the two key reproductive hormones in females. From the mammary gland to the ovary and from the uterus to the pituitary, these two hormones exert remarkable tissue- and cell-type specific actions. For example, while progesterone promotes mitogenic actions of E2 in the mammary gland, it inhibits estrogen-driven proliferation in uterine epithelia[9–12]. Given the indispensable role of the uterus in establishing pregnancy, cellular responses to E2 and P4 in the uterus are well studied during early pregnancy events[13,14].

Uterine epithelial and stromal cells must be ready to proliferate and then differentiate in response to the hormones E2 and P4 to prepare for embryo implantation[13]. Much of our understanding of the hormone-driven endometrial changes during pregnancy comes from rodent studies[15]. In early pregnant mice, pre-ovulatory estrogen promotes uterine epithelial proliferation. Then, progesterone allows stromal proliferation to continue but causes epithelial proliferation to stop, and the uterus becomes conducive for embryo attachment on day four of pregnancy[13,16]. In response to embryo attachment to the uterine epithelium, the underlying stromal cells stop proliferating and differentiate into decidual cells, which permit trophoblast invasion and placentation[17–19].

Similar steps occur in humans, though they occur in a cyclical pattern every ~28 days. During the proliferative phase of the menstrual cycle, estrogen promotes epithelial proliferation. Then, ovulation induces the production of progesterone, which promotes stromal proliferation followed by decidualization during the secretory phase of the menstrual cycle. If the oocyte is not fertilized, then menstruation occurs to shed the thickened endometrium[20,21]. However, if the uterus fails to respond appropriately to these hormones, the woman can experience recurrent pregnancy loss[22,23]. Further, if these tissues proliferate inappropriately after entering the peritoneal space via retrograde menstruation, a woman can develop endometriosis, a painful disease that often leads to infertility. Thus, to improve pregnancy outcomes and treat or prevent endometriosis, we must define the molecular pathways that govern both the uterine epithelial and stromal responses to steroid hormones.

One gene that may participate in both physiologically normal menstrual cycle/pregnancy-mediated changes to the endometrium and in pathological changes that occur in endometriosis is Growth Regulation by Estrogen in Breast Cancer 1 (GREB1). GREB1 was originally identified as an early prototypical estrogen-responsive gene that promotes estrogen-dependent proliferation of breast cancer

cells[24–26] and acts as a chromatin-bound estrogen receptor cofactor in these cells. Importantly, recent genome-wide studies found several genetic variants near the *GREB1* region in women with endometriosis[27,28]. However, the precise roles of GREB1 in normal endometrial physiology during pregnancy and pathology during endometriosis have not been determined.

Here, we report that mice lacking *Greb1* have severe subfertility due to impaired uterine responses to steroid hormones, specifically P4 actions. Additionally, we discovered that GREB1 mediates P4-mediated responses in the endometrium to promote embryo implantation via acting as a cofactor of PR. On the other side, we found that GREB1 promotes estrogen-driven endometriosis progression in an in vivo mouse model and promotes the proliferation of human endometriotic stromal and epithelial cells in vitro by functioning as an ER cofactor. In summary, our findings revealed a distinct feedforward mechanism between GREB1 and PR/ER-α that dictate hormone-dependent action in endometrial physiology and pathophysiology.

## Results

### GREB1 acts as a PR cofactor to govern P4-mediated transcription

To explore the precise role of GREB1 in endometrial function and dysfunction, we first investigated its role in normal endometrial physiology. Toward this, we first sought to determine whether GREB1 is expressed in normal human endometrial tissue. Analysis of GREB1 expression in endometrial biopsies from healthy women showed that both the proliferative and the secretory phases of the menstrual cycle have abundant GREB1 nuclear puncta in the glandular epithelial and stromal cells of the endometrium (Fig. 1a). However, compared to proliferative phase biopsies, secretory phase biopsies appeared to have higher levels of GREB1 expression in stromal cells (Fig. 1a).

Given the abundant GREB1 expression in the human endometrium, we wondered whether GREB1 expression was altered in women with recurrent implantation failure. To determine that, we examined published raw transcriptome data from endometrial biopsies from 10 women with recurrent implantation failure and 10 subfertile women with no such history[29]. Interestingly, those with recurrent implantation failure had significantly fewer *GREB1* transcripts than those without this history (Fig. 1b). Since progesterone controls the secretory phase of the menstrual cycle and the higher stromal GREB1 expression in the endometrial stroma during the secretory phase, we first investigated the impact of progesterone on GREB1 expression. To test this, we

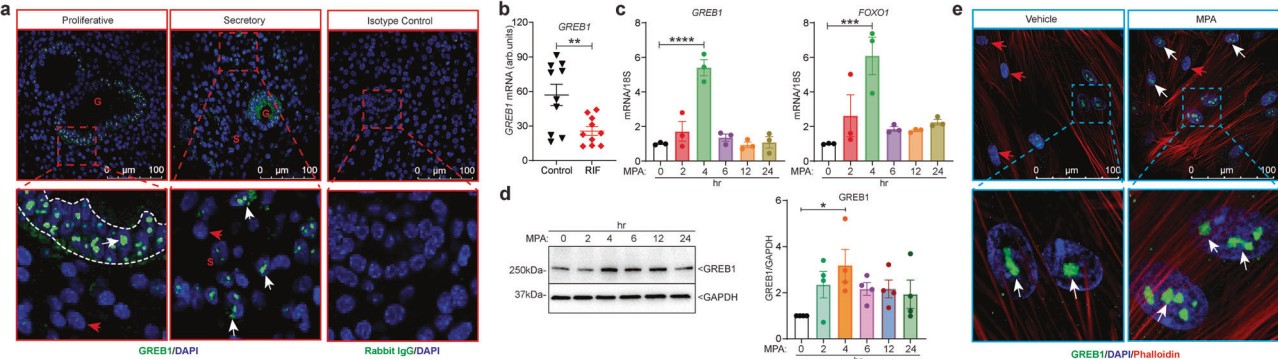

**Fig. 1 | GREB1 expression in the human endometrium is regulated by steroid hormones. a** Representative images of GREB1 staining in human endometrium from the proliferative ($n = 9$) and secretory ($n = 9$) phase. White arrow, GREB1-positive cells; red arrow, GREB1-negative cells. Right panel represents staining with the isotype control Rabbit IgG. **b** *GREB1* raw transcript scores in mid-secretory phase endometrium from women with and without recurrent implantation failure (RIF) measured from a publicly available GEO data set (GSE65102); ($n = 10$). Paired, two-tailed, t-test. Data reported as the mean ± SEM. *$P < 0.05$, ***$P < 0.001$, ****$P < 0.0001$. Relative amounts of *GREB1* and *FOXO1* mRNA (**c**), and GREB1 protein

(**d**) in human endometrial stromal cells treated with 1 μM MPA for the indicated number of hours. Analyzed by one-way ANOVA with Tukey's multiple comparisons post-test. Data reported as the mean ± SEM. *$P < 0.05$, ***$P < 0.001$, ****$P < 0.0001$. **e** Representative GREB1 immunofluorescence in human endometrial stromal cells treated with 1 μM MPA for 4 hr. Blue, DAPI; Green, GREB1; and Red, Phalloidin, data are reported as the mean ± SEM from three biological replicates from a representative experiment (experiment repeated three times). *$P < 0.05$, ***$P < 0.001$, ****$P < 0.0001$.

isolated primary human endometrial stromal cells (HESCs) and treated them with Progestin (MPA) for different time points. Within four hours of treatment, Progestin was observed to elevate both the GREB1 transcript (Fig. 1c) as well as protein levels (Fig. 1d) in HESCs, implying that GREB1 may be an early progesterone-responsive gene. As anticipated, the well-established progesterone-responsive gene Forkhead box protein O1 (*FOXO1*) transcript[30] was also induced in Progestin-treated cells (Fig. 1c). Similarly, immunofluorescence revealed that Progestin treatment increased the number of GREB1 puncta in the nuclei of HESCs (Fig. 1e) further corroborating our findings that induced expression of GREB1 in human endometrial stroma is mediated by progesterone. As nuclear-puncta are the active sites for transcription[31], we wondered whether GREB1 participates in steroid hormone-driven transcriptional regulation.

Thus, to determine whether GREB1 has any role in the progesterone-mediated transcription, we depleted GREB1 levels in HESCs and treated with progestin (MPA). Results showed that GREB1 knockdown significantly impaired induction of *FOXO1* expression in response to MPA (Fig. 2a) but had no effect on the expression of either the A or B isoforms of progesterone receptor (PR) (Fig. 2b). Given that GREB1 acts as an estrogen receptor coactivator in breast cancer cells, we postulated that GREB1 functions as a PR cofactor in the endometrium. To test this, we used a publicly available database of GREB1 and PR cistromes in MCF-7 breast cancer cells[32] to look for potential PR and GREB1 binding sites in the *FOXO1* gene. We found overlapping sites of potential GREB1 and PR binding at two enhancer regions within the *FOXO1* gene locus (Supplementary Fig. 1a). To determine whether GREB1 and PR occupy these sites in HESCs, we treated HESCs with MPA and then performed chromatin immunoprecipitation followed by region-specific PCR. Whereas PR occupied all four regulatory sites tested, GREB1 only occupied two. Neither protein occupied an untranscribed region (Fig. 2c). Importantly, knockdown of GREB1 in HESCs reduced the occupancy of PR on the *FOXO1* gene (Fig. 2d). Finally, co-immunoprecipitation assays showed the physical interaction between GREB1 and PR proteins in progestin treated HESCs (Fig. 2e). These data suggest that GREB1 associates with PR and functions as a cofactor, regulating P4-responsive gene expression. To further investigate this, we conducted Cut&Run sequencing in HESCs to identify the GREB1 cistrome. Our analysis identified over 2011 genomic regions bound by GREB1 in HESCs (Fig. 2f). Subsequent gene ontology analysis of GREB1-bound genes revealed significant enrichment in functional terms associated with transcription regulation, translation regulation, and DNA repair (Supplementary Fig. 1b). The visualization of binding profiles using IGV for GREB1 peaks on selected genes revealed a high enrichment of GREB1 on chromatin (Fig. 2g). Having identified a distinct cistrome for GREB1 in HESCs, we compared GREB1 peaks with the PR peaks generated by Dr. Demayo's group through Cut&Run sequencing in similar HESCs[33]. We identified that almost 50% of GREB1 binding regions in HESCs were also co-occupied by PR (Fig. 2h). Furthermore, a comparison of genes called from generated peaks revealed that about 63% of GREB1-bound genes were also occupied by PR (Fig. 2h). These findings suggest GREB1 acts as PR cofactor to target subset of P4/PR responsive genes to govern specific biological processes in endometrium.

## Loss of GREB1 impairs female fertility in mice

Although our in vitro studies found GREB1 as a key mediator of P4 actions, the precise role of GREB1 in endometrial responses to E2 and P4 can only be evaluated in in vivo mouse models. Thus, to dissect the in vivo role of GREB1 in endometrium, we first assessed whether the mouse uterus expresses GREB1 during the preparation of embryo implantation. In early pregnant murine uteri (from day 1 to 3 dpc), punctate GREB1 staining was observed in luminal and glandular epithelial cells, with only a few GREB1-positive stromal cells in stroma. However, beginning at 4 dpc, a time wherein uterus become receptive,

GREB1 expression start to elevate in stromal cells (Fig. 3a–c). Consistent with an idea that GREB1 staining correlates with decidualization, we detected more GREB1-positive stromal cells at implantation sites than at inter-implantation sites at 5 dpc (Fig. 3c). At 6 dpc, GREB1 was evident in the primary decidual zone, a transient avascular zone that initially protects the embryo as it implants (Fig. 3b). We detected little to no GREB1 expression in fully differentiated decidual polygonal cells (Fig. 3b). These findings collectively imply that GREB1 expression is increased during implantation and decidualization in both the endometrial epithelium and stroma compartment, and is subsequently downregulated once these processes are complete, indicating the relevance of GREB1 in these early pregnancy events.

Given the distinct spatiotemporal expression of GREB1 in murine uterus, we next investigated the in vivo function of GREB1 by generating *Greb1* knockout (KO) mice by using CRISPR/Cas9 to delete exons 10 through 17 (Fig. 4a, b). Homozygous *Greb1* KO mice were born at expected Mendelian ratios without any overt developmental defects other than a minor runted phenotype. However, *Greb1* KO pups developed normally, and the runted phenotype was not obvious by eight weeks of age. *Greb1* KO mice had no obvious developmental defects in the uterus, though we confirmed that neither GREB1 transcript nor protein were expressed in uteri from 8-week-old virgin *Greb1* KO mice (Fig. 4c, d). To assess the effects on female fertility, age-matched WT and *Greb1* KO females were mated to fertility-proven WT males for six months. *Greb1* KO females produced significantly fewer pups per litter (Fig. 4e) and fewer total pups per mouse (Fig. 4f) than WT females. Additionally, we also found that the *Greb1* KO females delivered reduced number of litters in an inconsistent manner over the testing period compared to controls (Supplementary Fig. 2a). To determine the cause of subfertility in *Greb1* KO females, we first assessed ovarian function. Immunohistochemistry analysis showed that GREB1 was present in all cell types of the ovaries in WT mice but was absent in the ovaries of *Greb1* KO mice (Supplementary Fig. 2b). Furthermore, histological examination of ovaries from adult females showed no overt morphological differences between *Greb1* KO and WT mice, both showing the presence of corpus lutea and normal follicular development (Fig. 4g). Additionally, we noted similar ovarian expression of Estrogen Receptor-α (ER-α) and PR at transcripts and protein level in *Greb1* KO and WT littermate mice (Supplementary Fig. 2c, d). We also assessed ovarian function by super-ovulating four-week-old mice and collecting oocytes. The oocytes from *Greb1* KO mice were able to be fertilized in vitro at a similar rate as oocytes from WT littermates (Fig. 4h). Finally, we mated *Greb1* KO and WT littermates and, at 4 dpc, recovered similar numbers of blastocysts (Fig. 4i). These findings suggest that the subfertility in *Greb1* KO mice is not due to impaired ovarian function or impaired oviductal transport of embryos.

## GREB1 is required for embryo implantation and uterine receptivity

Given the above findings, we hypothesized that the subfertility of *Greb1* KO mice might be due to impaired embryo implantation and/or uterine receptivity. To test this idea, we mated mice and then examined their uteri at 5 dpc, finding that *Greb1* KO mice had significantly fewer implantation sites than WT littermates (Fig. 5a). Additionally, in histological analysis, we observed small implantation chambers, incomplete luminal closure, and embryo misorientation in *Greb1* KO mice (Fig. 5b). Furthermore, *Greb1* KO mice appeared to have less stromal cell proliferation than WT littermates at both 4 dpc and 5 dpc (Fig. 5c, e). However, the uteri from *Greb1* KO mice had similar levels of estrogen receptor (*Esr1*) and progesterone receptor (*Pgr*) mRNA as of uteri from WT littermates at 5 dpc (Fig. 5d). Additionally, the uterine epithelia from WT and *Greb1* KO mice had similar levels of Mucin 1 at 4 dpc, a marker of uterine receptivity (Fig. 5f). Consistently, transcript levels of *Muc1* in uterine tissues from 4 dpc were unaltered in WT and

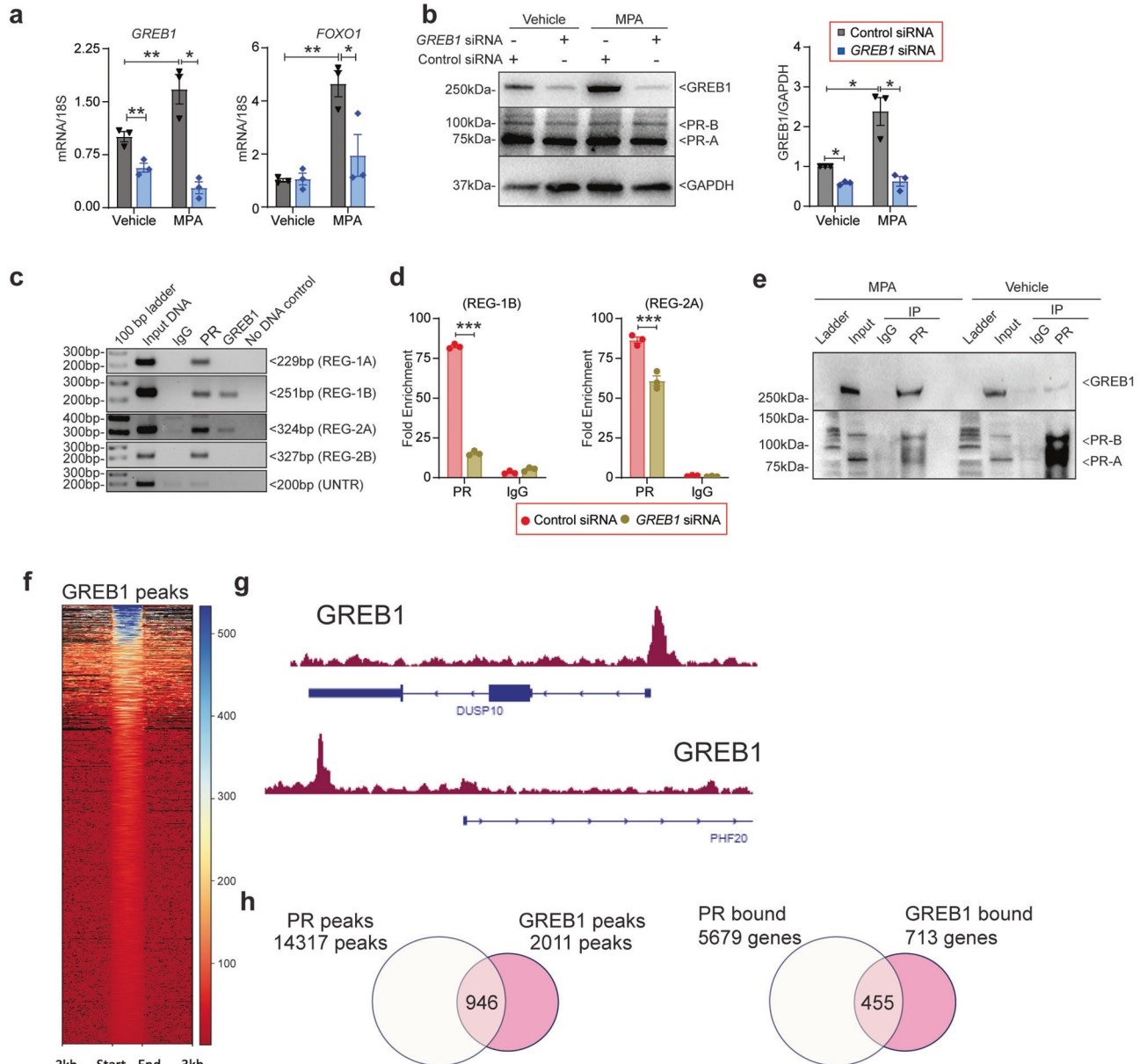

**Fig. 2 | GREB1 acts as a PR cofactor in human endometrial stromal cells.**
**a** Relative *GREB1* and *FOXO1* mRNA abundance in human endometrial stromal cells transfected with control or *GREB1* siRNA and treated with MPA or vehicle for 4 hr. Analyzed by one-way ANOVA with Tukey's multiple comparisons post-test. Data reported as the mean ± SEM. *$P < 0.05$, ***$P < 0.001$, ****$P < 0.0001$. **b** Relative GREB1 and PR protein concentrations in human endometrial stromal cells transfected with control or *GREB1* siRNA and treated with vehicle or MPA. Right side panel depicts the GREB1 protein quantification. GAPDH serves as a loading control. Analyzed by one-way ANOVA with Tukey's multiple comparisons post-test. Data reported as the mean ± SEM. *$P < 0.05$, ***$P < 0.001$, ****$P < 0.0001$. **c** PCR products amplified with the indicated *FOXO1* primers (shown in map in Supplementary Fig 1a, after immunoprecipitating DNA with anti-GREB1 or anti-PR antibody. UNTR, untranscribed region. **d** ChIP-qPCR validation of PR binding on the FOXO1 region in HESCs

treated with control siRNA or *GREB1* siRNA prior to 4 hr MPA treatment. Data are represented as fold enrichment of IgG and PR over that of the negative control region. Analyzed by one-way ANOVA with Tukey's multiple comparisons post-test. Data reported as the mean ± SEM. *$P < 0.05$, ***$P < 0.001$, ****$P < 0.0001$. **e** Whole-cell lysates isolated from human endometrial stromal cells treated with 1 µM MPA or vehicle for 4 h, immunoprecipitated with PR antibody or control IgG, and immunoblotted with GREB1 (top panel) or PR (bottom panel) antibody. Data are presented as the mean ± SEM from three biological replicates from a representative experiment (experiment repeated three times). *$P < 0.05$, **$P < 0.01$, ***$P < 0.001$. **f** The heatmap of GREB1 binding peaks from HESCs using CUT&RUN sequencing analysis. **g** IGV track visualization of GREB1 peaks on DUSP10 and PHF20 gene clusters. **h** Venn diagrams illustrating the overlap between GREB1 and PR peaks (left) and called genes (right) from the GREB1 and PR cistromes.

*Greb1* KO mice (Supplementary Fig. 3a). Together, these data suggest that loss of GREB1 impairs uterine stromal proliferation which results in impaired uterine receptivity that is necessary for embryo implantation.

Since the cell-type-specific proliferation and differentiation programs in the endometrium are strictly regulated by steroid hormones[34], we examined the effect of loss of *Greb1* on expression of

estrogen- and progesterone-responsive genes in uteri from mice at 4 dpc. Unexpectedly, we did not observe any change in the estrogen-regulated genes including Cyclin D1 (*Ccnd1*), Insulin-like growth factor 1 (*Igf1*), Fibroblast growth factor 18 (*Fgf18*) (Fig. 5g) and *Pgr* (Supplementary Fig. 3b) in the uteri from *Greb1* KO compared to control littermate mice. Conversely, we found that the expression of progesterone-responsive genes including Indian Hedgehog (*Ihh*),

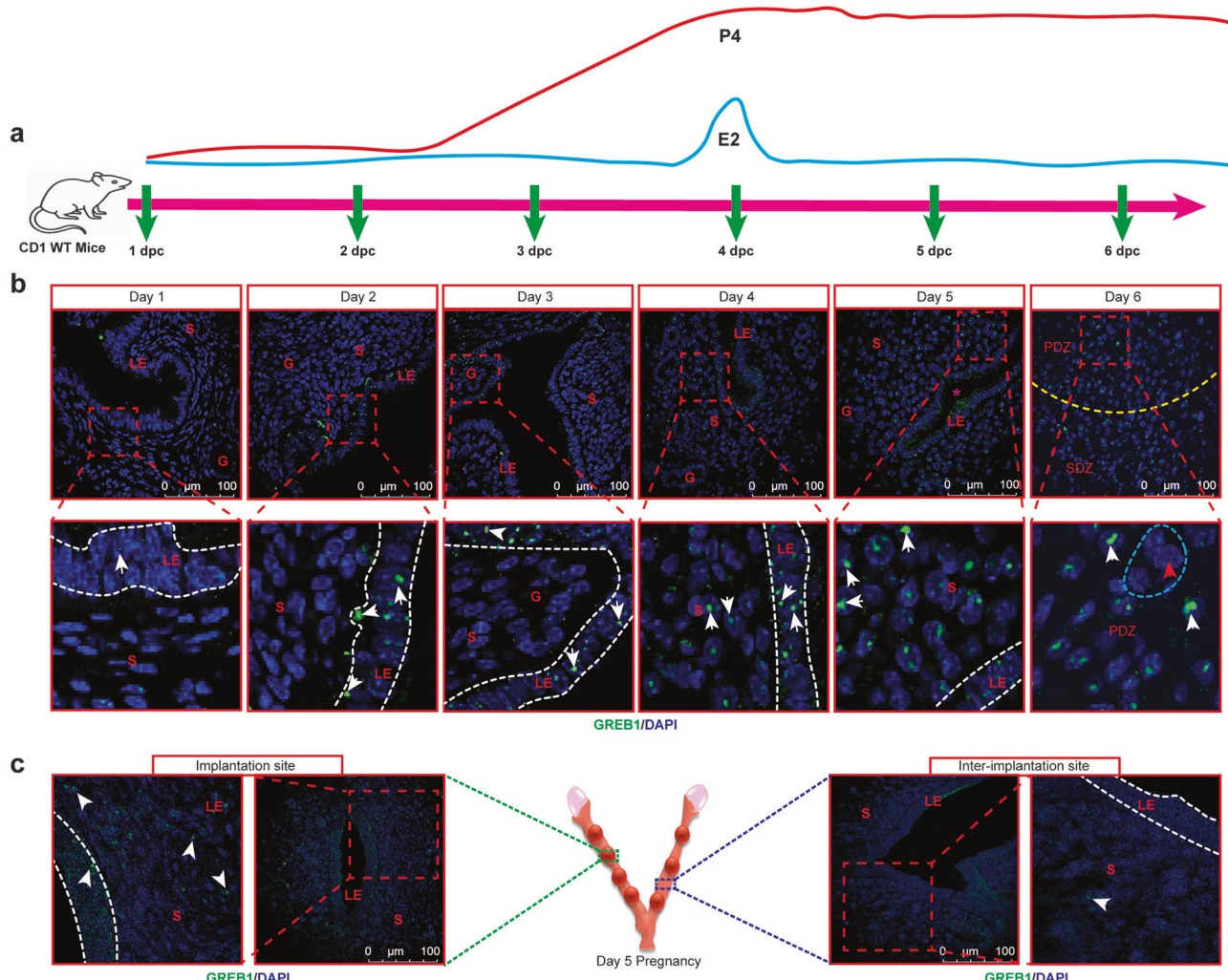

**Fig. 3 | GREB1 expression in the murine endometrium increases at the time of embryo implantation. a** Schematic representation of collection of uteri at different days of pregnancy (days post-coitum) in mice. **b** Representative images of GREB1 (green) localization in wild-type CD1 mouse uteri at the indicated days of pregnancy. Asterisk indicates the location of a blastocyst. G gland, LE luminal epithelium, S stroma, PDZ Primary decidual zone, SDZ Secondary Decidual zone; Scale bar:100 μm. White arrows, GREB1-positive cells; Red arrows, GREB1-negative cells; Blue dashed outline, decidual polygonal cells. **c** Representative images of GREB1 (green) localization at an implantation site and an inter-implantation site in a uterus from a wild-type CD1 mouse at 5 days. Middle schema (created with BioRender.com) depicts the implantation site and inter-implantation sites. In each time point, at least 5 independent samples (n = 5) from different mice were examined. All uteri were collected at 9:00 a.m. to 10:00 a.m. on the indicated day of pregnancy.

Interleukin 13 Receptor Subunit Alpha 2 (*Il13ra2*), Cytochrome P450 Family 26 Subfamily 1 (*Cyp26a1*), and *Foxo1* (Fig. 5h and Supplementary Fig. 3c), were much lower in the uteri from *Greb1* KO mice compared to control counterparts. However, we did not find significant difference in expression of Amphiregulin (*Areg*) and *Hand2* as shown in Supplementary Fig. 3c. These findings imply that loss of GREB1 impairs uterine receptivity that precedes embryo implantation by impeding progesterone-responses but not estrogen action in endometrium.

## GREB1 mediates P4-action but not E2-responses in normal endometrium

To dissect the precise role of GREB1 in uterine responses to estrogen and progesterone, we used a well-established controlled steroid hormone regimen (delayed implantation model) to artificially induce uterine receptivity. In this model, ovariectomized mice are treated with one of three regimens. In "estrogen priming", mice receive two days of estrogen treatment, rest for two days, then receive vehicle for four days. In "estrogen group", mice receive two days of estrogen treatment, rest for two days, receive vehicle for three days, and then receive estrogen for 16 h. Finally, in "estrogen/progesterone", mice

receive two days of estrogen treatment, rest for two days, receive progesterone for three days, and then receive estrogen plus progesterone for 16 h (Fig. 6a). Mice in this last group will develop a uterus that is receptive to embryo implantation[35,36]. Immunostaining for the proliferation marker phospho-Histone H3 (PH3) demonstrated that *Greb1* KO and WT littermates had similar endometrial epithelial proliferation in the estrogen-treated group (Fig. 6b middle panel). Consistent with this finding, loss of *Greb1* did not affect expression of the estrogen-responsive genes *Igf1*, *Mcm2*, *Klf4*, or *Klf15* in these models (Supplementary Fig. 4a–d). However, in the estrogen/progesterone groups, uteri from *Greb1* KO mice had significantly fewer proliferating (PH3-positive) cells than did uteri from WT littermates (Fig. 6b, c), indicating that GREB1 mediates specific progesterone responses in stromal cells but not required for estrogen-mediated epithelial cell proliferation.

Since progesterone-dependent endometrial stromal cell proliferation precedes endometrial decidualization, we wondered whether loss of *Greb1* would impair decidualization. Thus, we used an artificial decidualization model in which mice are ovariectomized and then one uterine horn is injected with sesame oil to induce

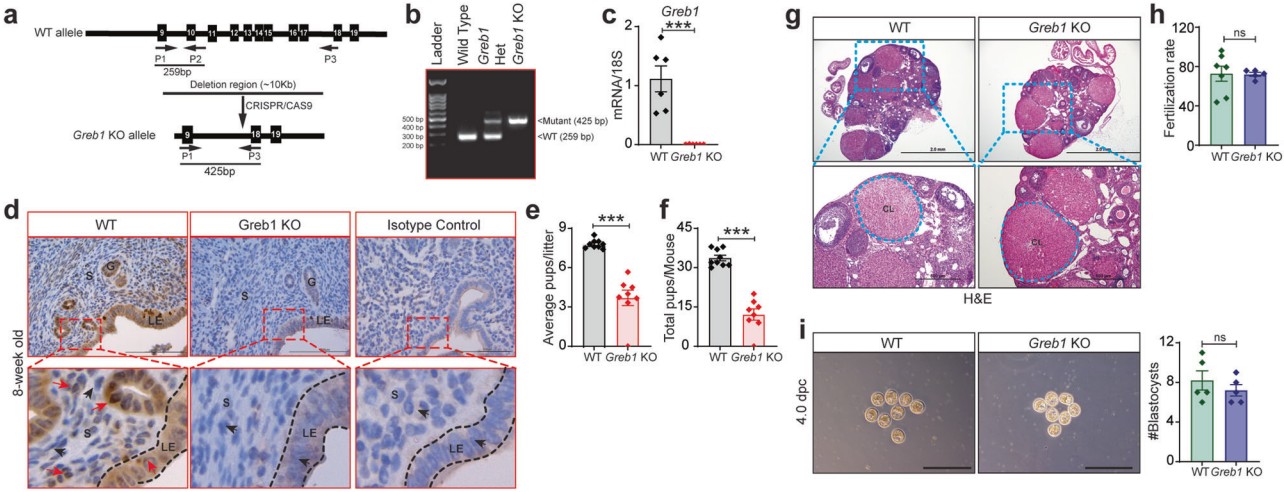

**Fig. 4 | *Greb1* knockout (*Greb1* KO) mice have impaired fertility. a** Schematic of *Greb1* knockout strategy. **b** Confirmation of wild type (WT) and mutant alleles by PCR. Representative image of PCR, observed in at least five specimens from different mice **c** Analysis of *Greb1* transcripts in uteri from WT (*n* = 6) and *Greb1* KO mice (*n* = 6). Paired, two-tailed, t-test. Data reported as the mean ± SEM. *$P < 0.05$, ***$P < 0.001$, ****$P < 0.0001$. **d** Representative cross-sectional images of uteri from WT and *Greb1* KO mice stained for GREB1 by immunohistochemistry. Scale bar, 200 μm; Red arrow, GREB1-positive cells; black arrow, GREB1-negative cells. G, glandular epithelia; LE, luminal epithelia; S, stroma. Representative image of at least three specimens analyzed per genotype. **e, f**, Graphs depicting the number of pups per litter and the total number of pups per mice from WT (n = 9) and Greb1 KO (*n* = 8) in

six-month fertility tests. Paired, two-tailed, t-test. Data reported as the mean ± SEM. *$P < 0.05$, ***$P < 0.001$, ****$P < 0.0001$. **g** Representative (*n* = 5) Hematoxylin and Eosin-stained cross-section images of the ovary from 8-week-old WT and *Greb1* KO mice; scale bar: 2.0 mm and 200 μm. CL corpus luteum. Representative image of at least five specimens analyzed per genotype. **h** In vitro fertilization rate of oocytes recovered from 4-week-old WT (*n* = 7) and *Greb1* KO mice (n = 5). Paired, two-tailed, t-test. Data reported as the mean ± SEM. *$P < 0.05$, ***$P < 0.001$, ****$P < 0.0001$. **i** Representative images and quantification of blastocysts (graph on right) retrieved from WT (*n* = 5) and *Greb1* KO (*n* = 5) mice on 4 dpc. Paired, two-tailed, t-test. Data are presented as mean ± SEM, (*n* = 5-9 WT and n = 5-8 *Greb1* KO). ***$P < 0.001$ and ns, non-significant.

decidualization (Supplementary Fig. 5a). The stimulated uterine horn in WT mice enlarged significantly more and had more proliferative cells than did the stimulated uterine horn in *Greb1 KO* mice (Supplementary Fig. 5b, c). Moreover, expression of the decidualization markers bone morphogenetic protein 2 (*Bmp2*) and wingless-related mouse mammary tumor virus integration site 4 (*Wnt4*) increased significantly more in stimulated uterine horns from WT mice than in those from *Greb1* KO mice (Supplementary Fig. 5d). Together, these results suggest that GREB1 plays an important role in progesterone-dependent endometrial stromal cell proliferation and decidualization. Further, these findings are consistent with our previous finding that GREB1 is required for human endometrial stromal cell decidualization[30].

To determine whether reduced stromal cell proliferation in *Greb1* KO mice was due to altered early progesterone transcriptional responses, we ovariectomized mice, subcutaneously injected them with progesterone or vehicle (oil), and then analyzed gene expression in their uteri[37]. Progesterone treatment induced *Greb1* mRNA expression in uteri from WT mice, indicating that *Greb1* is an early PR target gene (Fig. 6d). However, *Pgr* expression was not affected by progesterone treatment in uteri from WT or *Greb1* KO mice (Fig. 6d). The progesterone-regulated genes *Areg, Ihh*, and *Cyp26a1* were less upregulated in uteri from *Greb1* KO mice than in those from WT mice (Fig. 6e). In contrast, another progesterone-regulated gene *Il13ra2* was upregulated by progesterone to a similar extent in uteri from *Greb1* KO and WT mice (Fig. 6e, lower right panel). Together, these results suggest that progesterone promotes GREB1 expression, and then GREB1 works with PR to regulate expression of selective genes in a feed-forward mechanism to govern uterine receptivity.

### GREB1 promotes estrogen-dependent action in endometriosis

Given that GREB1 mediates estrogen actions in breast malignancies, we were surprised by the lack of GREB1 impact on estrogenic action in normal endometrium and wondered whether this is the case in

endometrial pathologies as well. To explore this, we investigated the role of GREB1 in one of the prominent endometrial pathology, endometriosis. We chose to focus on endometriosis as this is estrogen-driven pathology and GREB1 SNPs were reported in this genealogical condition. To dissect this, we used a mouse model of endometriosis in which a piece of the uterus is autologously transplanted onto the peritoneum, resulting in growth of lesions that resemble human endometriosis in numerous respects[38]. In WT mice and human endometriotic lesion, immunohistochemical and immunofluorescence analysis revealed that GREB1 was more abundantly expressed in the ectopic lesions than in the control uterus/endometrium (Fig. 7a, b). Next, we used this same model to induce endometriosis in *Greb1* KO and WT littermate mice and found that G*reb1* KO mice developed smaller endometriotic lesions than did WT littermates (Fig. 7c–f). Moreover, whereas lesions from WT mice had a thick epithelial layer and expressed GREB1, lesions from *Greb1* KO mice had thinner epithelial layers and did not express GREB1 (Fig. 7g, h). Consistent with their larger size, lesions from WT mice had significantly more proliferative (Ki-67 positive and Cyclin D1 positive) epithelial and stromal cells than did lesions from *Greb1* KO mice (Fig. 7i–k and Fig. 8a, b). Further, analysis of serum estrogen levels revealed no significant differences between *Greb1* KO and control mice (Supplementary Fig. 6).

Finally, to confirm that GREB1 mediates estrogen-driven proliferation in human endometriotic cells, we used siRNA to knock down *GREB1* expression in an endometriotic epithelial cell line and in primary human endometriotic stromal cells and then treated the cells with vehicle or estrogen. Estrogen treatment significantly increased proliferation in cells that received control siRNA but not in cells that received *GREB1* siRNA (Fig. 8c, d). We analyzed expression of estrogen target genes in these cells and found that estrogen induced expression of *GREB1, CCND1*, and *IGF1* is significantly downregulated with *GREB1* knockdown (Fig. 8e). These findings suggest that GREB1 plays an important role in estrogen-driven endometriosis disease progression. Taken together, our findings suggest that through a similar feed-

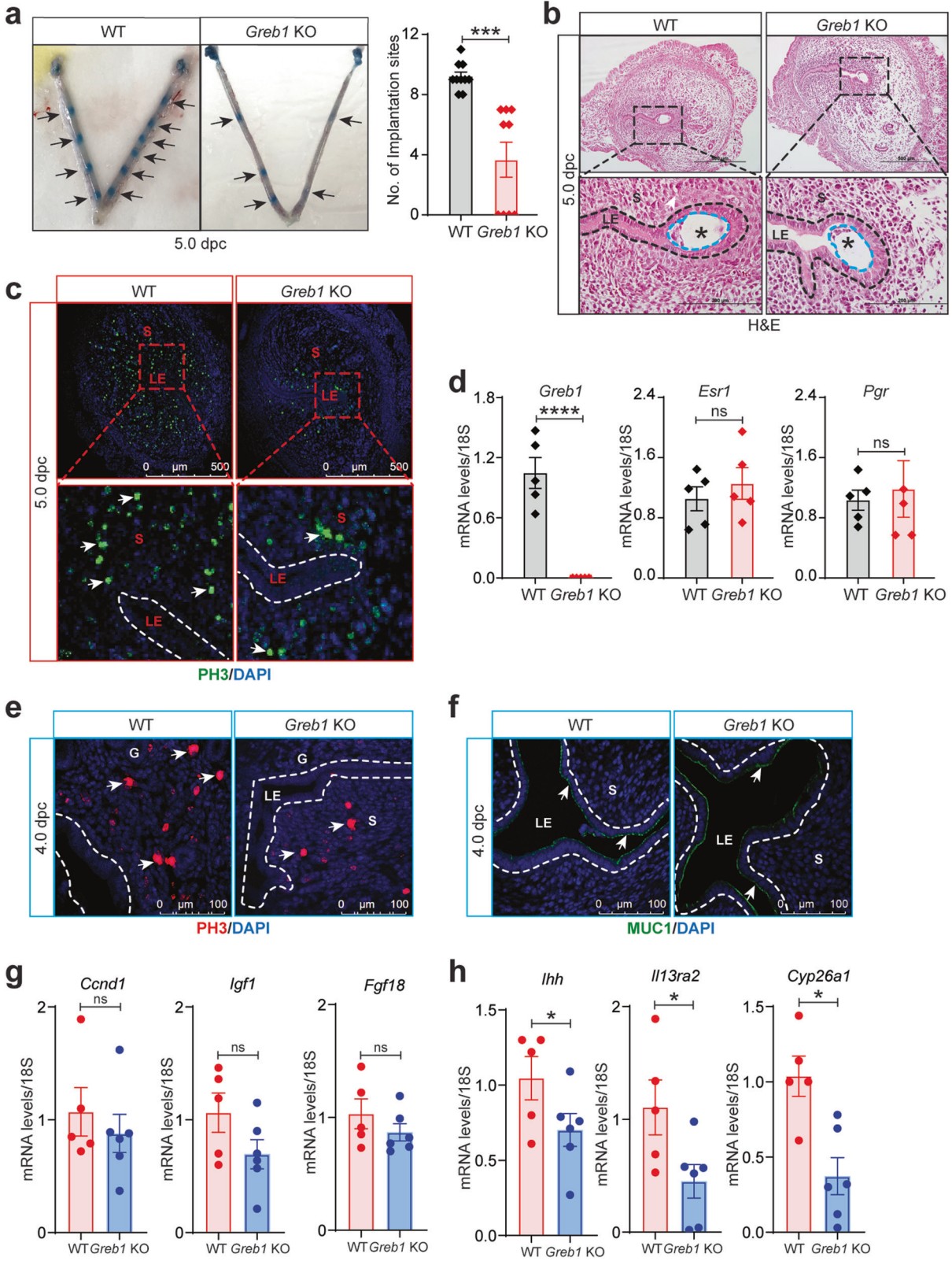

forward mechanism, GREB1 acts differentially on estrogen and progesterone responses based on normal or diseased state of endometrium. In normal receptive endometrium, GREB1 mediates progesterone functions by acting as a PR co-factor. Contrastingly, in a pathological state of endometrium, it mediates the effects of estrogen, likely acting as an ER co-factor.

## Discussion

The pleiotropic and cell-type specific actions of steroid hormones are governed by the differential expression of receptors and the coregulators[2–5]. Recent evidence found feed forward regulatory loops between transcription factors and other proteins govern multiple cellular functions[39,40]. However, such feed forward mechanisms are

**Fig. 5 | *Greb1* KO mice have impaired embryo implantation and uterine receptivity. a** Embryo implantation sites at 5 dpc in WT (*n* = 10) and *Greb1* KO (*n* = 11) mice. Black arrows indicate the implantation sites. Paired, two-tailed, t-test, data reported as the mean ± SEM. \**P* < 0.05, \*\*\**P* < 0.001, \*\*\*\**P* < 0.0001. **b** Representative Hematoxylin and Eosin-stained cross-section images of the uterus at 5 dpc in WT and *Greb1* KO mice; scale bars, 500 μm and 200 μm. Representative image of at least three specimens analyzed per genotype. **c** Representative cross-sectional images of uteri of WT and *Greb1* KO mice at 5 dpc stained for Phospho-Histone H3, scale bar 500 μm. White arrows indicate positive cells. Representative image of at least three specimens analyzed per genotype. **d** Relative mRNA

expression of indicated genes at 5 dpc in WT and *Greb1* KO females. Paired, two-tailed, t test, data reported as the mean ± SEM. \**P* < 0.05, \*\*\**P* < 0.001, \*\*\*\**P* < 0.0001 and ns, non-significant **e**, **f**, Representative cross-sectional images of WT and *Greb1* KO mice at 4 dpc uteri stained for Phospho-Histone H3 (**e**), and MUC1 (**f**) Representative image of at least three specimens analyzed per genotype. **g** Relative mRNA levels of estrogen target genes *Ccnd1*, *Igf1* and *Fgf18*, and **h** progesterone target genes *Ihh*, *Il13ra2* and *Cyp26a1*, in the uteri of *Greb1* KO and WT mice at dpc 4 (*n* = 5 for each genotype). Asterisks denote the blastocysts. LE, luminal epithelia; S, stroma. Scale bar, 100 μm. Paired, two-tailed, t-test, data are presented as mean ± SEM (*n* = 5–6 mice per group). \*\*\**P* < 0.001, \*\*\*\**P* < 0.0001, and ns, non-significant.

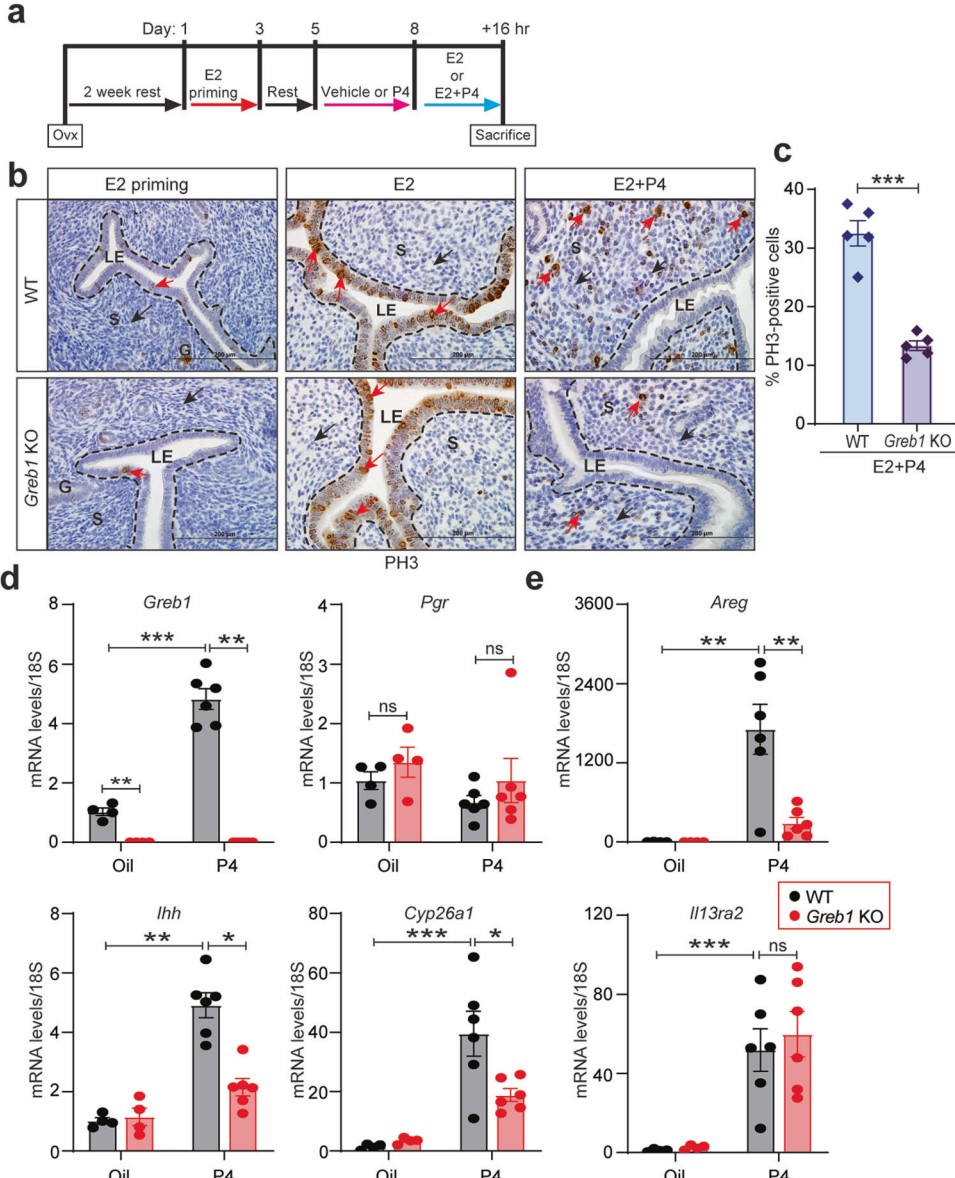

**Fig. 6 | GREB1 is required for P4-responses but not E2-responses in receptive endometrium. a** Experimental protocol for hormonal induction of uterine receptivity in ovariectomized mice. **b** Representative images of uteri from WT and *Greb1* KO mice from indicated groups, stained for phospho-Histone H3. G, glandular epithelia; LE luminal epithelia, S stroma. **c** The graph displays the percentage of phospho-Histone H3 positive endometrial stromal cells in WT and *Greb1* KO mice from the E2 + P4 group. Red arrow, PH3-positive cells; black arrow, PH3-negative

cells; *n* = 5 mice per group. Paired, two-tailed, t-test. Data are presented as mean ± SEM. \**P* < 0.05, \*\**P* < 0.01, \*\*\**P* < 0.001 and ns, non-significant Relative mRNA expression of *Greb1* and *Pgr* (**d**) and indicated PR target genes (**e**), in uteri from WT and *Greb1* KO mice in the indicated treatment groups, n = 5 mice per group. Analyzed by one-way ANOVA with Tukey's multiple comparisons post-test. Data are presented as mean ± SEM. \**P* < 0.05, \*\**P* < 0.01, \*\*\**P* < 0.001, and ns, non-significant.

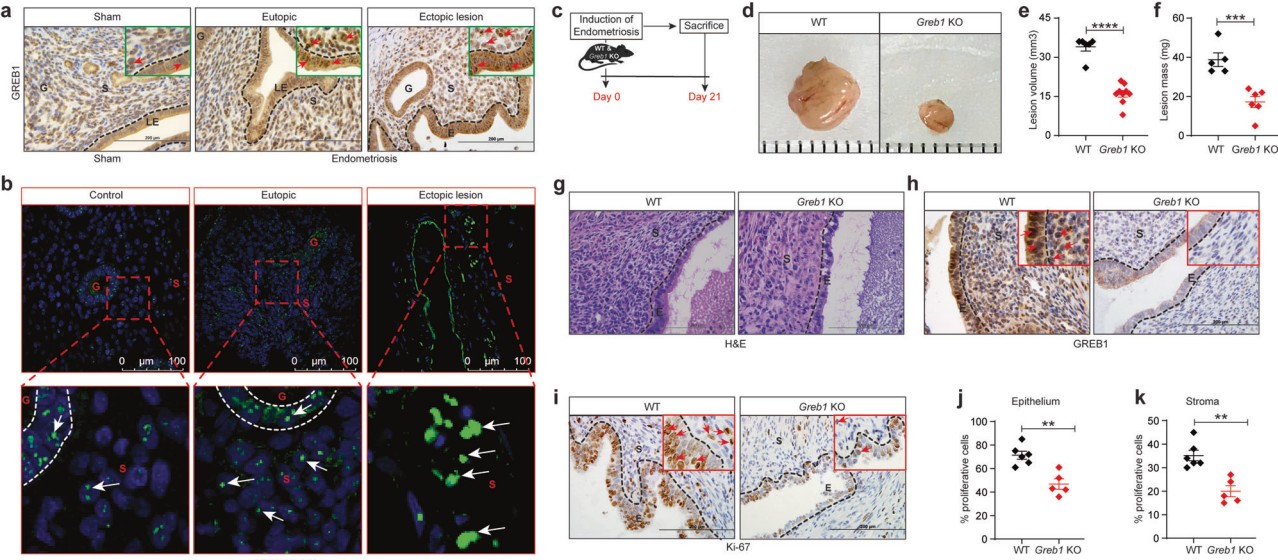

**Fig. 7 | GREB1 is required for endometriotic lesion growth in mice. a-b**, Representative images of GREB1 localization in mouse eutopic endometrium and ectopic lesion, (*n* = 5) (**a**) and human eutopic endometrium and ectopic lesion (*n* = 10 control, *n* = 10 eutopic and n = 10 ectopic lesion) (**b**). Red/White arrows, GREB1-positive cells. **c** Experimental timeline and procedure. Ectopic endometriotic lesion representative images (**d**), volumes WT (*n* = 6) and *Greb1* KO (*n* = 9) (**e**), and masses WT (n = 5) and *Greb1* KO (n = 6), **f** from mice 21 days after surgical induction of endometriosis. Paired, two-tailed, t-test. Data are presented as the mean ± SEM.

*P < 0.05, **P < 0.01, ***P < 0.001 and ns non-significant. Representative images of ectopic lesions from WT and *Greb1* KO mice stained with Hematoxylin and Eosin (**g**), anti-GREB1 antibody (**h**), and anti-Ki-67 antibody (**i**); red arrows, indicates respective positive cells (*n* = 5). Graphs display percentage of Ki-67-positive cells in endometriotic lesion epithelium (**j**), and stroma (**k**) from WT (*n* = 6) and *Greb1* KO mice (*n* = 5). E epithelium, G gland, LE luminal epithelium, S stroma. Paired, two-tailed, t-test. Data are presented as the mean ± SEM. *P < 0.05, **P < 0.01, ***P < 0.001 and ns non-significant.

well understood for either progesterone receptor, or estrogen receptor in mediating their ligand actions. In this study, we found a feed forward regulatory axis between GREB1 and these two receptors that govern the physiological or pathological actions of steroid hormones (Fig. 8f). Specifically, we show that, in both mouse and human endometrial cells, GREB1 expression is upregulated by progesterone which in turn, helps regulate progesterone-mediated gene expression. This function appears to be essential for endometrial decidualization and embryo implantation. Conversely, in both mouse and human endometriotic cells, GREB1 expression is regulated by estrogen, and GREB1 is important for estrogen-mediated gene expression. This work thus places GREB1 in a feedforward loop with progesterone receptor in normal endometrial physiology and in a feedforward loop with estrogen receptor in endometrial pathology.

Our findings may have relevance to early pregnancy loss, in which conception occurs but embryo demise arises before six weeks, the time of implantation in healthy pregnancies. Although the majority of early pregnancy losses (which occur in 30–60% of women)[41] are attributed to chromosomal abnormalities, some occur because of the non-receptive uterus and when the embryo fails to implant[19,42,43]. Our observations of severe subfertility and impaired decidualization in *Greb1* KO mice suggest that GREB1 has an essential role in uterine function during early pregnancy. In humans, this idea is supported by both our previous finding that loss of *GREB1* impaired in vitro decidualization[30] and our current finding that endometrial *GREB1* mRNA expression was lower in women with recurrent implantation failures than in women without such a history. Future efforts should be directed at determining whether measuring endometrial GREB1 expression could be used as a means of diagnosing or predicting early pregnancy loss.

We noted an important difference between physiological and pathological regulation and function of GREB1. Whereas GREB1 was required for progesterone-mediated effects in normal endometrial physiology of pregnancy, it was not required for estrogen effects. In contrast, GREB1 was controlled by estrogen and contributed to estrogen effects in endometriosis. Our findings are consistent with the

role of GREB1 in mediating estrogen-induced proliferation and androgen-induced proliferation in breast and prostate cancer cells respectively[25,44]. Thus, GREB1 appears to be a pan-steroid hormone response mediator that functions in signalling in response to progesterone, estrogen, and androgen receptors. Although we found GREB1 is not required for estrogen-mediated action in uterine epithelia, we cannot rule out the possibility of GREB1 mediated epithelial-stromal paracrine signaling in mediating the endometrial functions. To dissect this, studies must be carried out on epithelial- and stromal-specific conditional knockout mouse models.

The pleiotropic effects of steroid hormones are orchestrated through their cognate receptors and fine-tuned by coactivators and cofactors. We identified a distinct GREB1 cistrome, revealing that half of the regions where GREB1 binds are also occupied by PR and more than 60% of GREB1 bound genes are occupied by PR. Moreover, GREB1 did not bound on all the genes bound by PR, implying GREB1 as one of the key cofactor of PR, but not the only cofactor. Interestingly, much of the ER binding sites on the chromatin are also bound by GREB1 and depletion of ER reduced the GREB1 occupancy on the chromatin[32]. Further, recent evidence revealed that GREB1 catalyses O-GlcNAcylation of ER-α, which is required for stabilization of ER-α protein by inhibiting association with the ubiquitin ligase[45]. We found GREB1 acts as a PR cofactor in endometrial stromal cells and is not required for ER function in normal endometrium and interestingly O-GlcNAcylation of PR regulates the functions in breast cancer[46]. This intricate interplay between GREB1/PR/ER likely plays a role in the pleiotropic effects of steroid hormones, highlighting the complexity of gene regulation and cellular responses to hormonal signals. Therefore, one important question for future work is whether GREB1 has context- or tissue-specific interactions with hormone receptors. For example, GREB1-PR and GREB1-ER-α interactions on chromatin might together coordinate the gene expression changes required for endometrial receptivity. As a future prospective it will be interesting to see whether GREB1 plays any role in glycosylation and stabilization of PR during endometrial function.

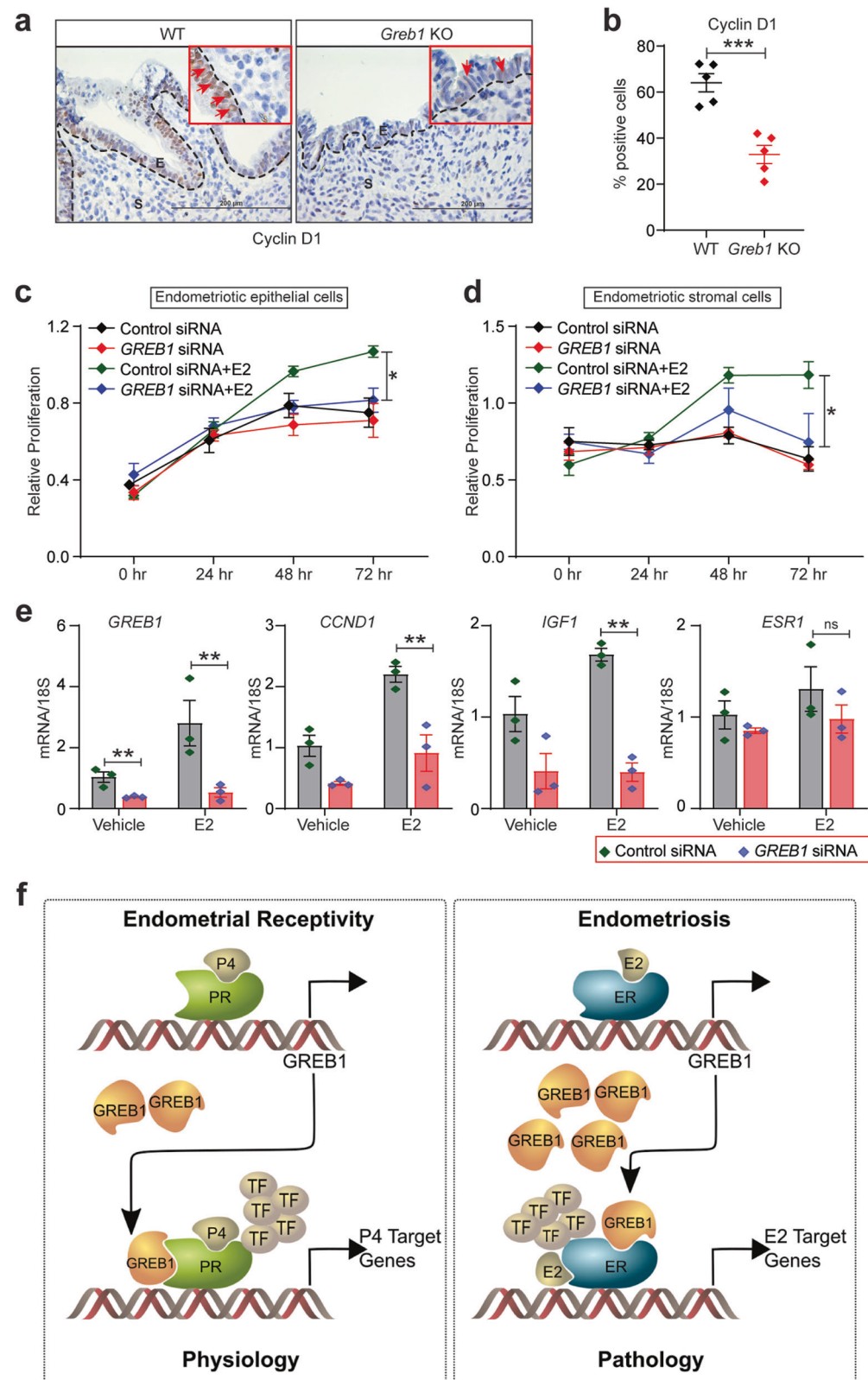

The Genome-Wide Association Studies (GWAS) have documented Single Nucleotide Polymorphisms (SNPs) in patients with endometriosis, predominantly in the stage III/IV cases. In different sample populations, there has been replication of SNPs near genes involved in estrogen and other steroid regulated pathways including *ESR1* and *GREB1*[47]. The multiple SNPs of GREB1 constitutes most consistently associated gene with endometriosis population[27,28,47–50]. Considering the reported SNPs of GREB1 in endometriosis[27,28,47–50], it is intriguing to test the impact of these SNPs on GREB1 interaction with steroid hormone receptors. Examining the functional relevance of these individual SNPs of GREB1 in endometriosis disease progression is of our immediate future focus. Moreover, our work highlights the need for studies to identify the transcription factors that function with GREB1 to promote overall female reproductive health.

**Fig. 8 | GREB1 is required for estrogen-dependent action in endometriosis.**
**a** Representative images of ectopic lesions from wild type and *Greb1* KO (n = 5) mice stained with anti-Cyclin D1. Red arrow, Cyclin D1-positive cells. E epithelium, S stroma. **b** Graph displays percentage of Cyclin D1-positive cells in endometriotic lesion epithelium. Paired, two-tailed, t-test. Data are presented as mean ± SEM. *P < 0.05, **P < 0.01, ***P < 0.001 and ns, non-significant. **c-d** Representative MTT proliferation assays of Immortalized Human Endometriotic Epithelial Cells. **c**, and primary stromal cells isolated from human endometriotic lesions (HEnSCs) **d** from the indicated groups and time points. Data are presented as the mean ± SEM from triplicate samples from one experiment (three experiments were conducted in

total). *P < 0.05, **P < 0.01, ***P < 0.001 and ns, non-significant. **e** Relative abundance of *GREB1, CCND1, IGF1*, and *ESR1* transcripts in Human Endometriotic Epithelial Cells transfected with control or *GREB1* siRNA and treated with estrogen or vehicle for 6 h. Data are presented as the mean ± SEM from triplicate samples from one experiment (three experiments were conducted in total). Analyzed by one-way ANOVA with Tukey's multiple comparisons post-test, *P < 0.05, **P < 0.01, ***P < 0.001 and ns non-significant. **f** Schematic illustration (created with in association with InPrint at Washington University in St. Louis) of the hypothesis that GREB1 participates in both endometrial physiology and pathology.

## Methods

### Ethical approvals
Human endometrial tissues were obtained from participants under a protocol approved by the Washington University in St. Louis School of Medicine Institutional Review Board (IRB ID # 201612127 and 201807160). The study was conducted in accordance with the criteria set by the Declaration of Helsinki. All participants were recruited through the Washington University online classified section and local newspaper advertisements. Eligible participants signed an Informed Consent and Authorization form. All animal studies were performed according to a protocol approved by the Institutional Animal Care and Use Committee of Washington University School of Medicine, Saint Louis, MO, USA (protocol number 20160227).

### Study design
The objective of this study was to determine the role of GREB1 in the endometrium during early pregnancy and endometriosis. First, GREB1 expression was assessed in the human endometrium during two phases of the menstrual cycle and in response to progesterone in human endometrial stromal cells in vitro. Second, the potential association of GREB1 and PR on chromatin was assessed. Third, the role of GREB1 in pregnancy was assessed by analyzing fertility, implantation, and decidualization in *Greb1* KO and WT littermate mice. Fourth, the requirement for GREB1 in estrogen-dependent endometriotic lesion growth was assessed in a surgical model of endometriosis in *Greb1* KO and WT littermate mice. Finally, whether GREB1 is required for estrogen-dependent proliferation of endometriotic epithelial and stromal cells was assessed in vitro. The numbers of biological and technical replicates (n) for in vivo and in vitro studies for each experiment are mentioned in the respective figure legends. For each experiment, the sample size (n) and numbers of technical replicates were determined by the investigators on the basis of pilot studies or experience with standard disease models. Animals were assigned to control and treatment groups in an unbiased manner and housed together to minimize experimental differences arising from environmental effects. All endpoints were assessed such that the investigator was blinded to treatment group or genotype, as relevant for each type of experiment.

### Animal husbandry
All transgenic mice were maintained on a C57BL/6 genetic background (The Jackson Laboratory, Bar Harbor, ME) to minimize variation in the gestation length. All experimental animals were housed 5 per cage in institutional animal facility in standard ventilated cages with free access to water and food and under a 12-r light and dark cycle. Cages were changed routinely, and the health of the mice was monitored daily, and only healthy mice were used for this study. Breeding was carried out in duos or trios.

### Collection of human samples and human endometrial stromal cell isolation
Potential participants were excluded if they had used probiotics, antibiotics, or any anti-inflammatory drugs within two weeks before

surgery or had a history of uterine fibroids, polycystic ovarian syndrome, or endometrial cancer. Human endometrial stromal cells were isolated as described previously[51]. Briefly, endometrial biopsies were trimmed into small pieces using sterile scissors and subsequently digested in DMEM/F12 medium containing collagenase (2.5 mg/ml (Sigma-Aldrich) and DNase I (0.5 mg/ml (Sigma-Aldrich) for 1.5 h at 37 °C. Following digestion, dispersed cells were collected by centrifugation and layered over a Ficoll-Paque reagent layer (GE Healthcare Biosciences, Pittsburgh, PA) to remove lymphocytes. The top layer containing the hESC fraction was collected and filtered through a 40 µm nylon cell strainer (BD Biosciences, Franklin Lakes, NJ) to separate the epithelial cells. Fractionated HESCs or HEnSCs were then resuspended in DMEM/F-12 media containing 10% FBS, 100 units/ml penicillin and 0.1 mg/ml streptomycin (HESCs OR HEnSCs media) and cultured in tissue culture flasks (75 cm²). All experiments with human endometrial stromal cells were performed independently with three technical replicates of cells derived from three independent patients. Ectopic endometriotic lesions and eutopic endometrial biopsies were collected from women undergoing endometriosis surgery and proceeded for stromal cell isolation as described above.

### Immunofluorescence of uterine tissue and human endometrial stromal cells
After tissue collection, mice uterine tissues and human endometrial biopsies were processed for immunofluorescence staining. Prior to staining, tissues were fixed in 4% paraformaldehyde and subsequently embedded in paraffin. Tissue sections (5 µM) underwent deparaffinization in xylene, followed by rehydration in an ethanol gradient, and antigen retrieval by boiling in citrate buffer (Vector Laboratories Inc., CA, USA). Following blocking with 2.5% normal goat serum in phosphate-buffered saline (PBS) from Vector Laboratories for 1 h at room temperature, sections were incubated overnight at 4 °C with primary antibodies (see Supplementary Table 2). After rinsing with PBS, sections were treated with Alexa Fluor 488 or 546-conjugated secondary antibodies from Life Technologies for 1 h at room temperature, followed by washing and mounting with ProLong Gold Antifade Mountant with DAPI from Thermo Scientific.

Similarly, human endometrial stromal cells were cultured on poly-Lysine coated coverslips (Sigma-Aldrich) in 12-well plates. Upon reaching 80–90% confluence, cells were treated with 1 µM MPA for 4 h, followed by PBS washing and fixation with 4% PFA in PBS for 15 min at room temperature. Subsequently, cells were permeabilized with 0.2% Triton X-100 (Sigma Aldrich, USA) in PBS for 10 min at room temperature, and then processed for blocking and staining as described for tissue sections above.

### Real-time qPCR
Cells or tissues were lysed in lysis buffer, and total RNA was isolated with the Purelink RNA mini kit (Invitrogen, Carlsbad, CA, USA) according to the manufacturer's instructions. RNA was quantified with a Nano-Drop 2000 (Thermo Scientific, Waltham, MA, USA). Then, 1 µg of RNA was reverse transcribed with the High-Capacity cDNA Reverse Transcription Kit (Thermo Scientific, Waltham, MA, USA). The

amplified cDNA was diluted to 10 ng/µL, and QPCR was performed with primers specified in Supplementary Table 1 and Fast TaqMan 2X Mastermix (Applied Biosystems/Life Technologies, Grand Island, NY) on a 7500 Fast Real-time PCR system (Applied Biosystems/Life Technologies, Grand Island, NY). The delta-delta cycle threshold method was used to normalize expression to the reference gene 18 S.

## Western blotting

Protein lysates (40 µg per lane) were loaded on a 4–15% SDS-PAGE gel (Bio-Rad), separated in 1× Tris-Glycine Buffer (Bio-Rad), and transferred to PVDF membranes via a wet electro-blotting system (Bio-Rad), all according to the manufacturer's instructions[52]. PVDF membranes were blocked for 1 h in 5% non-fat milk in Tris-buffered saline containing 0.1% Tween-20 (TBS-T, Bio-Rad), then incubated overnight at 4 °C with antibodies listed in Supplementary Table 2 in 5% bovine serum albumin (BSA) in TBS-T. Blots were then probed with anti-Rabbit IgG conjugated with horseradish peroxidase (1:3000, Cell Signaling Technology) in 5% BSA in TBS-T for 1 h at room temperature. Signal was detected with the Immobilon Western Chemiluminescent HRP Substrate (Millipore, MA, USA), and blot images were collected with a Bio-Rad ChemiDoc imaging system.

## siRNA transfection

Human endometrial stromal cells, human endometriotic stromal cells or immortalized human endometriotic epithelial cells with expressing luciferase (IHEECs/Luc) were plated in six-well culture plates and treated in triplicate with Lipofectamine 2000 reagent (Invitrogen Corporation, Carlsbad, USA) and 60 pmol of the following siRNAs: non-targeting siRNA (D-001810-10-05) or siRNAs targeting *GREB1* (L-008187-01-0005) (GE Healthcare Dharmacon Inc., Lafayette, CO) as described previously[30]. After 48 h, cells were treated with 1 µM MPA, 100 nM E2 (Sigma-Aldrich), or ethanol as a vehicle in 1× Opti-MEM-I reduced-serum media (Invitrogen Corporation, Carlsbad, USA) with 2% Charcoal Stripped-Fetal Bovine serum (cs-FBS).

## Analysis of chromatin immunoprecipitation (ChIP)-seq datasets, and PCR validation

Publicly available ChIP-seq data for PR and GREB1 chromatin occupancy in MCF-7 cells (GSE41561)[32] were analyzed. The two regulatory regions bound by both PR and GREB1 were visualized with the UCSC genome browser (Supplementary Fig. 1a). The ChIP assays were performed with the ChIP-IT Express kit (Active Motif Inc., Carlsbad, CA). Briefly, human endometrial stromal cells were cultured to 70–80% confluency in a 15 cm cell culture dish, treated with 1 µM MPA for 4 h, and then fixed with 10% formaldehyde for 10 min. Cells were then lysed and Dounce homogenized to obtain the nuclear fraction. Nuclei were suspended in chromatin shearing buffer and sonicated (Covaris, E200, Woburn, MA, USA) to fragment the chromatin to roughly 500–1000 bp fragments. Fragmented chromatin was immunoprecipitated overnight at 4 °C with 3 µg of rabbit polyclonal antibody specific to either human PR or human GREB1 (listed in Supplementary Table 2). Incubation with a rabbit polyclonal IgG antibody served as a control for non-specific immunoprecipitation (listed in Supplementary Table 2). Then, fragmented chromatin was reverse cross-linked, and the immunoprecipitated DNA fragments were eluted. Regions of the FOXO1 gene upstream region that were enriched for PR or GREB1 were identified by PCR employing specific primers (listed in Supplementary Table 3) and the following cycle parameters: (94 °C for 20 s, 59 °C for 30 s, 72 °C for 30 s) for 35 cycles. Amplification products were electrophoresed on 1% Agarose/1xTAE gels and stained with ethidium bromide. UNTR is from an untranscribed region[53].

## Chromatin immunoprecipitation-quantitative PCR (ChIP-qPCR)

Briefly, HESCs were cultured to 50–60% confluency in a 15 cm cell culture dish and treated with Lipofectamine 2000 reagent (Invitrogen Corporation, Carlsbad, USA) and following siRNAs: non-targeting siRNA (D-001810-10-05) or siRNAs targeting *GREB1* (L-008187-01-0005) (GE Healthcare Dharmacon Inc., Lafayette, CO) as described above. After 48 h of incubation these cells were treated with 1 µM MPA for 4 h, and then ChIP was carried out as mentioned in above section. For ChIP-qPCR input chromatin was used to generate a standard curve for the amplification of each primer set to determine the amount of DNA immunoprecipitated by IgG and PR antibodies. Binding data were represented as the fold enrichment over the input. Primers that were designed to span the Upstream regions of the FOXO1 gene that were enriched for PR were identified by PCR (listed in Supplementary Table 3).

## Immunoprecipitation

Protein extracts (1 mg) from human endometrial stromal cells treated with either vehicle (ethanol) of 1 µM MPA for 4 h were pre-cleared with normal Mouse IgG-Agarose. The cleared extracts were then incubated overnight with 5 µg anti-PR-Agarose or normal Mouse IgG-Agarose (listed in Supplementary Table 2) at 4 °C. Protein-bead complexes were captured by centrifugation at 14,000 × *g* for 15 min and washed with 1 ml of wash buffer (150 mM NaCl, 1 mM EDTA, 10 mM Tris and 0.1% Triton X-100) for 15 min. The centrifugation and washing steps were performed three times. Protein samples were resolved by 4–15% SDS-PAGE and subjected to immunoblotting with GREB1 and PR antibodies.

## CRISPR-mediated *Greb1* deletion

Two single guide RNAs (sgRNAs) were selected by the Baylor College of Medicine (BCM) Mouse Embryonic Stem Cell Core by using the Wellcome Trust Sanger Institute Genome Editing website (http://www.sanger.ac.uk/htgt/wge/). The sgRNAs were designed to flank the genomic region containing the open reading frame of *Greb1* exons 10 through 17. The sgRNAs chosen had at least 3 mismatches with genes other than *Greb1* (5′ sgRNA: https://www.sanger.ac.uk/htgt/wge/crispr/347586428 and 3′ sgRNA: https://www.sanger.ac.uk/htgt/wge/crispr/347585044). The sgRNAs were synthesized by the BCM mES Core from DNA templates produced via high-fidelity PCR[54] and purified with the QiaQuick PCR purification kit. The MEGAshortscript T7 kit (ThermoFisher, AM1354) was used for in vitro transcription. RNA was then purified with the MEGAclear Transcription Clean-Up Kit (ThermoFisher AM1908). All samples were analyzed by Nanodrop to determine concentration and visualized with the Qiaxcel Advanced System and the RNA QC V2.0 kit to check the RNA quality, then stored at −80 °C. Cas9 mRNA was purchased from ThermoFisher (A25640). All sgRNAs were reanalyzed by Nanodrop before assembling the microinjection mixtures, which consisted of Cas9 mRNA (100 ng/µL) and sgRNA (10 ng/µL each) in a final volume of 60 µL RNAse-free PBS.

## Microinjection of CRISPR/Cas9 reagents

C57BL/6NJ female mice, 24–32 days old, were intraperitoneally injected with 5 IU/mouse of pregnant mare serum gonadotropin (PMSG), followed 46.5 hr later with 5 IU/mouse of human chorionic gonadotropin (hCG). They were then mated to C57BL/6NJ males. Fertilized oocytes were collected at 0.5 days post-coital (dpc). The BCM Genetically Engineered Rodent Model Core microinjected the sgRNA/Cas9/ssOligo mixture into the cytoplasm of at least 100 pronuclear stage zygotes. Approximately 25–32 injected zygotes were transferred into each pseudo-pregnant ICR females (8–9-week-old) on the afternoon of the injection. [ICR females [Crl:CD1(ICR)] were purchased from Charles River (Raleigh, NC)].

## Genotyping

G0 mice were genotyped by standard PCR with three primers: a single forward primer (P1, GACAGGGTGTTTCCTTTTG) and two reverse primers (P2, TTAGGCCACCATTGGAAACT, and P3, ACAACCTCAGGCTGC AATTT). Amplification with P1 and P3, which were approximately 425

bases outside the two sgRNA sites, generated an amplicon in a CRISPR-mediated deletion allele only. Amplification with primers P1 and P2 only produced a product (approximately 259 bases) if the WT allele was present.

## Tissue collection from adult mice

Prior to collecting tissue, all adult females from wild type as well as *Greb1* KO were first examined for estrous cycle. Vaginal smears were obtained by flushing the vaginal opening with 25 μL of PBS and fluid containing cell suspension was spotted onto glass microscope slides. Slides were allowed to air-dry, stained with 0.1% crystal violet stain and examined under the brightfield microscope. The stage of the estrous cycle was determined based on the presence or absence of leukocytes, cornified epithelial, and nucleated epithelial cells. Most of the adult females euthanized for uteri or ovaries collection were from diestrus or metestrus stage of estrous cycle.

## CUT&RUN experiment

CUT&RUN was performed as per previously published protocol[55]. Briefly, endometrial stromal cells were overexpressed with GREB1 plasmid construct (Genecopoeia, catalogue number: EX-Y4970-M91) and then treated with MPA for 4 h. Overexpressed cells were collected by trypsin digestion and frozen viably in the freezing medium (90% FBS with 10% DMSO) until experiment day. A total of $10 \times 10^6$ cells were incubated with IgG, and GREB1 antibody and processed further following the CUT&RUN Assays (Epicypher) manual. Sequencing libraries were prepared using NEBNext Ultra II DNA Library Prep Kit (New England BioLabs, Cat #E7645) following manufacture's protocol. Paired end 150 bp sequencing was performed on a NEXTSeq550 (Illumina) platform and each sample was targeted for 10–15 million reads.

## CUT&RUN sequence alignment

The CUT&RUN pipeline was adapted from CUT & RUN Tools[56]. Raw fastq files were processed to remove adapters from sequenced reads using Trimmomatic v 0.39 (2:15:4:4: true LEADING:20 TRAILING:20 SLIDINGWINDOW:4:15 MINLEN:25) with the Truseq3.PE.fa adapter library, and then trimmed again using the kseq script[56,57]. Quality of trimmed reads was assessed using FastQC v0.11.9. Reads were aligned to both the hg38 (GENCODE GRCh38.p14 primary assembly release 43) and spike-in Ecoli K12 genomes (GCF_000005845.2_ASM584v2), using bowtie2 version 2.4.2 (--dovetail --phred33)[58]. BAM files of aligned reads were converted to BED files, and then bedgraphs, using BEDTools v2.30.0[59]. Blacklisted regions, from the hg38 blacklist downloaded from CUT & RUN Tools, were removed from the BED files. Each sample was normalized to its internal Ecoli spike-in using the spike_in_calibration.sh[60]. Spike-in normalized bedgraphs were converted to bigwigs using the UCSC bedGraphToBigWig utility[61]. Integrated Genome Viewer (IGV) v2.11.1 was used to examine spike-in normalized bigwig tracks at individual loci[62].

## CUT&RUN analysis

GREB1 peaks were identified using MACS2 with a p-adjusted (qValue) cutoff of 0.05[63]. PR peaks were identified in a previous study, also using MACS2 with a p-adjusted (qValue) cutoff of. 0.01[33]. Overlapping peaks were identified using BEDTools, with the overlap defined as 1 bp. Peak associated genes were identified as peaks within 5 kb of the TSS of a gene. BigWigs of individual replicates were merged using wiggletools and UCSC utilities wigToBigWig[64]. Merged bigwigs were used to generate a heatmap of CUT& RUN peaks with deeptools v 3.5.4[65]. Cut&Run sequencing data are available in the GEO database under accession number GSE254175.

## Fertility analysis

Female fertility was evaluated by individually pairing one 8-week-old *Greb1* KO (n = 8) and one WT littermate control (*Greb1*[+/+]) (n = 9) mouse with one WT male of proven fertility. The numbers of pups per litter and litters per mouse were tracked over six months for each female. Data are reported as mean ± SEM.

## Natural pregnancy studies in mice

Sexually mature 8–10-week-old female CD1 wild-type (for data in Fig. 3) or *Greb1* KO and WT littermates (for data in Fig. 4, Fig. 5, and Supplementary Fig. 3) were mated with fertility-proven WT males overnight. Copulation was confirmed by the observation of vaginal plugs on the following morning, which was designated as 1 dpc. Mice were euthanized, and uteri were collected on 1, 2, 3, 4, 5, and 6 dpc from CD1 WT and at 4 dpc or 5 dpc from *Greb1* KO and WT littermates. For mice sacrificed on 4 dpc and 5 dpc, the numbers of implantation sites and recovered blastocysts were counted for each genotype. To measure the ovulation rate, blastocysts were flushed from the uterine horns with PBS at 4 dpc (just before implantation) and counted. Data are reported as mean ± SEM.

## Embryo implantation studies

On the 5th day post-coitum (dpc), mice were administered a 100 μL tail-vein injection of 1% Chicago Blue dissolved in PBS and filtered through a 40 μm filter. Approximately 5 min following the Chicago Blue dye injection, the mice were euthanized. Subsequently, dissected uteri were photographed, and the blue bands indicating implantation sites were counted. Data are presented as mean ± standard error of the mean (SEM).

## In vitro fertilization

Four-week-old female mice were intraperitoneally injected with 5 IU PMSG (Sigma-Aldrich, St. Louis, MO, USA), followed by 5 IU hCG (Sigma-Aldrich) 46–48 h later. On the next day, sperm were collected from WT mice in Toyoda, Yokoyama and Hosi media (CytoSpring, CA, USA) supplemented with 4 mg/ml BSA (Sigma-Aldrich) and incubated at 37 °C in 5% $CO_2$. The mice were euthanized, and oocytes were collected from the oviduct in IVF media (RVF cook's media (Cook Medical, Inc. Bloomington, IN, USA) with 100 mM reduced Glutathione (Sigma-Aldrich) and incubated at 37 °C in 5% $CO_2$. After 1 h, 7–10 μl of sperm suspension was transferred into the IVF media containing oocytes and incubated at 37 °C in 5% $CO_2$ for 4–6 h. The oocytes were then washed and incubated in cleavage media (RVF cook's media with 0.5 mM EDTA) overnight at 37 °C in humidified condition with 5% $CO_2$. The fertilization rates were calculated by dividing the number of two-cell stage embryos by the number of oocytes.

## Controlled hormone regimen to mimic the hormonal states of pregnancy

Using a previously described method[35,36], *Greb1* knockout (KO) and wild-type (WT) littermates underwent bilateral ovariectomy under ketamine anesthesia with buprenorphine-SR as analgesics. After a two-week resting period to allow for dissipation of endogenous ovarian hormones, mice were injected with 100 ng of estrogen (E2; Sigma-Aldrich) dissolved in 100 μL of sesame oil over two consecutive days, followed by two days of rest. Subsequently, mice were randomly assigned to three groups with n = 5 mice in each group: "E2 priming" mice received four consecutive days of sesame oil injections, "E2" mice received three days of sesame oil injections followed by a single injection of 50 ng of E2 on the fourth day, and "E2 + P4" mice received 1 mg of progesterone (P4; Sigma-Aldrich) for three consecutive days followed by a single injection of 1 mg P4 plus 50 ng E2 on the fourth day. Hormones were delivered via subcutaneous injection in a 9:1 ratio of sesame oil to ethanol. Mice were euthanized 16 h after the final hormone injection to collect uteri. A portion of tissue from one uterine horn was processed in 4% neutral buffered paraformaldehyde for histology, while the remaining tissue was snap-frozen and stored at −80 °C until processed for RNA isolation.

## Artificial induction of decidualization

Artificial decidualization was induced following established protocols[35,66,67]. Briefly, six-week-old female *Greb1* knockout (KO) and wild-type (WT) littermates underwent bilateral ovariectomy under ketamine anesthesia with buprenorphine-SR as analgesics. After a two-week resting period to allow for dissipation of endogenous ovarian hormones, mice were subcutaneously injected with 100 ng of estrogen (E2) for three consecutive days. Following two days of rest, mice received subcutaneous injections of 1 mg of progesterone (P4) and 10 ng of E2 for three consecutive days. Six hours after the third E2 + P4 injection, 50 μL of sesame oil was injected into the lumen of the right uterine horn, while the untreated left uterine horn served as a control/unstimulated horn. Mice received subcutaneous injections of 1 mg P4 and 10 ng E2 for two additional days. Six hours after the last injection, mice were euthanized, and the wet weights of both uterine horns were recorded. Tissue from both uterine horns was fixed in 4% neutral buffered paraformaldehyde, snap-frozen, and stored at -80 °C until processed for RNA isolation.

## Assessment of progesterone target genes in mice

Six-week-old *Greb1* KO and WT littermates were bilaterally ovariectomized. After resting for two weeks to allow the endogenous ovarian-derived steroid hormones to dissipate, mice were subcutaneously injected with 100 μL sesame oil (vehicle control) or 1 mg progesterone (Sigma-Aldrich, St. Louis, MO) dissolved in 100 μL sesame oil[68]. After 6 h, mice were euthanized; uterine horns were fixed in 4% neutral buffered paraformaldehyde, snap-frozen, and stored at -80 °C until processed for RNA isolation. Total RNA was subjected to qPCR to assess P4-target genes as shown in Fig. 6d, e.

## Induction of endometriosis

Endometriosis was induced by auto-transplanting the uterine tissue from estrus-stage mice onto the peritoneal wall, as described previously[38,69,70], with minor modifications. Briefly, one uterine horn from 10-week-old, estrus-stage *Greb1* KO and WT littermates was excised and slit longitudinally under ketamine anaesthesia and buprenorphine-SR as analgesics. With the help of a dermal biopsy punch, a 3-mm endometrial fragment was removed and sutured onto the peritoneal wall in the same mouse through a midline incision[38,71–74]. For the sham surgery, a thread was sutured onto the peritoneal wall in a similar procedure. After 21 days, the lesions were collected, measured, weighed, fixed in 4% neutral buffered paraformaldehyde, snap-frozen, and stored at −80 °C. Before the sacrifice, blood was also collected, and serum was separated from the blood by centrifugation and stored at −80 °C before hormone analysis. Serum E2 levels were measured by using ELISA kits (Enzo life Sciences) according to the manufacturer's instructions (for data in Supplementary Fig. 6).

## Histological analysis

Uterine sections were processed for immunostaining following established protocols[38]. Briefly, after deparaffinization, sections underwent rehydration in an ethanol gradient and were then subjected to antigen retrieval by boiling for 20 min in citrate buffer (Vector Laboratories Inc., CA, USA). Endogenous peroxidase activity was blocked with BLOXALL (Vector Laboratories Inc., CA, USA), and tissue sections were blocked with 2.5% normal goat serum in PBS for 1 h (Vector Laboratories Inc., CA, USA). After three washes in PBS, sections were incubated overnight at 4 °C in 2.5% normal goat serum containing the primary antibodies listed in Supplementary Table 2. Following incubation, sections were treated with biotinylated secondary antibody for 1 h, washed, and incubated with ABC reagent (Vector Laboratories Inc., CA, USA) for 45 min. Color development was achieved with 3, 3'-diaminobenzidine peroxidase substrates (Vector Laboratories Inc.), and sections were counter-stained with Hematoxylin. Finally, sections were dehydrated and mounted in Permount

histological mounting medium (Fisher Scientific). Two independent investigators, blinded to experimental groups, counted positive cells in images taken at ×400 magnification. Cells were counted in at least four areas of each tissue section, and the percentage of positive cells relative to the total number of cells was calculated as described previously[35]. For hematoxylin and eosin (H&E) staining, uterine, ovarian, and endometriotic lesion tissue sections were fixed, processed, embedded, and sectioned as described above. After deparaffinization, sections were stained with hematoxylin and eosin following established protocols[52].

## Cell proliferation assay

Cell proliferation was assessed using the MTT assay (Promega, Madison, WI, USA) as per the manufacturer's instructions. The endometriotic epithelial cell line IHEECs/Luc and primary human endometriotic stromal cells (HEnSCs) were transfected with Lipofectamine 2000 reagent (Invitrogen Corporation, Carlsbad, CA, USA) along with either control siRNA (D-001810-10-05) or GREB1 siRNA (L-008187-01-0005). Subsequently, 48 h post-transfection, $5 \times 10^3$ IHEECs/Luc cells and $3 \times 10^3$ HEnSCs were seeded per well of a 96-well plate. After overnight incubation in 1X Opti-MEM supplemented with 2% cs-FBS, the cells were treated with 100 nM estrogen or vehicle (ethanol) in 1× Opti-MEM media containing 2% cs-FBS. The relative proliferation rate was determined using the MTT proliferation kit at 0, 24, 48, and 72 h.

## Statistics

All data are presented as mean ± SEM. A two-tailed paired student's t-test was used to analyze experiments with two experimental groups, and a two-way ANOVA was used to analyze experiments containing more than two groups. GraphPad Prism 7.03 software was used for all statistical analyses. $P < 0.05$ was considered significant. $*P < 0.05$, $**P < 0.01$, $***P < 0.001$, and $****P < 0.0001$.

## Reporting summary

Further information on research design is available in the Nature Portfolio Reporting Summary linked to this article.

# Data availability

The authors declare that all data supporting the findings of this study are available within the article and its supplementary information files or from the corresponding author upon reasonable request. Cut&Run-sequencing datasets generated in this study have been deposited in the NCBI GEO database under accession code: GSE254175. Source data are provided with this paper.

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

## Acknowledgements

We thank Dr. Deborah J. Frank (Department of Obstetrics and Gynecology, Washington University) for assistance with manuscript editing. Illustration for Fig. 8f was prepared with assistance by Dr. Dan Murphy, and Dr. Anushree Seth, in association with InPrint at Washington University in St. Louis. Uterus graphical illustration used in Fig. 3 was downloaded from BioRender.com and modified with adobe illustrator software. We also thanks Dr. Kexin Ku, University of Texas Health Science Center, San Antonio, Texas, USA for providing the GREB1 antibody. We thank Ashirbad Guria, (Department of Pathology & Immunology, Baylor College of Medicine, Houston, Texas) for assisting with Cut & Run experiment. This work was funded, in part, by National Institutes of Health (NIH)/National Institute of Child Health and Human Development (NICHD) grants: R01HD104813, R01HD102680, R01HD065435 and R00HD080742 to RK; R01 HD07857 to BWO; R01 HD042311 to J.P.L.; HD105800 to D.M., partly supported by the Intramural Research Program of the National Institute of Environmental Health Sciences, Z1AES103311 to F.J.D. and by the United States Department of Agriculture (USDA/ARS) under Cooperative Agreement No. 58-3092-0-001 to H.K.Y. and partly supported by the Genetically Engineered Rodent Model Core at BCM. The GERM Core is funded in part by the National Institutes of Health Cancer Center Grant (P30 CA125123).

## Author contributions

SBC and RK designed experiments, conducted most of the studies, analysed the data, and wrote the manuscript. PP, EA, ZL, and CEF conducted some of the experiments, and RBL, MD, TW, and HKY performed bioinformatics analysis. DGL and JDH generated Greb1 null mice. SJG, PTJ, KHM, JPL, ESJ, DM, FJD, and BWO provided reagents and reviewed the final draft of the manuscript. RK conceived the project and supervised the work. All authors critically reviewed the manuscript.

## Competing interests

## Additional information

[1]Department of Pathology and Immunology, Baylor College of Medicine, One Baylor Plaza, Houston, TX 77030, USA. [2]Department of Obstetrics and Gynecology, Center for Reproductive Health Sciences, Washington University School of Medicine, St. Louis, MO 63110, USA. [3]Department of Pediatrics, Baylor College of Medicine, One Baylor Plaza, Houston, TX 77030, USA. [4]Integrative Bioinformatics, National Institute of Environmental Health Sciences, Research Triangle Park, NC, USA. [5]Department of Molecular and Human Genetics, Baylor College of Medicine, One Baylor Plaza, Houston, TX 77030, USA. [6]Department of Molecular and Cellular Biology, Baylor College of Medicine, One Baylor Plaza, Houston, TX 77030, USA. [7]Lester and Sue Smith Breast Center, Baylor College of Medicine, One Baylor Plaza, Houston, TX 77030, USA. [8]Reproductive and Developmental Biology Laboratory, National Institute of Environmental Health Sciences, Research Triangle Park, NC, USA. [9]USDA/ARS Children's Nutrition Research Center, Department of Pediatrics, Baylor College of Medicine, One Baylor Plaza, Houston, TX 77030, USA. [10]Jan and Dan Duncan Neurological Research Institute, Texas Children's Hospital, Houston, TX 77030, USA. [11]Department of Obstetrics and Gynecology, Fienberg School of Medicine, Chicago, IL 77030, USA. [12]Department of Molecular Virology and Microbiology, Baylor College of Medicine, One Baylor Plaza, Houston, TX 77030, USA. ✉e-mail: Rama.Kommagani@bcm.edu

