## [Peer Review File · Nature Communications]

A distinct feed-forward mechanism between GREB1 and steroid receptors dictates hormone-dependent GREB1 action in endometrial function and dysfunctionREVIEWER COMMENTS

Reviewer #1 (Remarks to the Author):

Gaining a better understanding of the underlying mechanisms coordinating progesterone and estrogen receptor signaling in the reproductive tract is vital to improving women's reproductive health. The manuscript by Chadchan et al. utilizes a combination of in vitro and in vivo models to investigate the role of GREB1 in regulating endometrial steroid hormone signaling in normal and pathological conditions. The authors demonstrate interesting alterations in GREB1 function between physiological and pathological conditions. Specifically, the manuscript indicates that the deletion of GREB1 results in subfertility stemming from a decreased progesterone response. In comparison, GREB1 is required for estrogen-driven proliferation in a model of endometriosis. Overall, this is a scientifically sound manuscript that will interest reproductive biologists and clinical scientists. The few points I have are detailed below.

1. Line 183. A published report suggests that human endometrial epithelial organoids are responsive to estrogen (PMID: 31666317). The authors may consider the RNA-seq data from that publication.
2. Line 189-190. Progesterone does not induce decidualization in the mouse. The sentence needs to be rephrased for accuracy.
3. Line 121- Replace infertility with subfertility
4. The subfertility phenotype warrants further descriptive data from the 6-month breeding trial. (number of litters per animal and time between litters)
5. The authors present data indicating no difference in the number of blastocysts recovered from WT and GREB1 KO mice (fig 4), but there is a significant difference in implantation site number (figure 5). In that regard, it is unclear if the embryo orientation defect found on GD 5 (figure5) is observed in the implantation sites demarcated by Chicago blue dye or in all embryo sections.
6. Line 247- Lif is an E2-responsive gene on gestational day 4.
7. In the endometriosis model, it is essential that the authors validate that there is no difference in circulating E2 levels between global GREB1 KO and WT mice. The reviewer understands that serum may not have been saved from this experiment and believes it would be acceptable to determine circulating estrogen levels in a separate cohort of intact mice at estrus.

Reviewer #2 (Remarks to the Author):

The cell type specific responses to the steroid hormones and the intricate interactions between ER and PR still remain elusive in female reproduction with little progress have made in recent decade. The manuscript entitled "A distinct feed-forward mechanism between GREB1 and steroid receptors dictates hormone dependent GREB1 action in endometrial function and dysfunction" describe the pathophysiologic function of GREB1 in female endometrium. The authors proposed that GREB1 mainly mediate progesterone activity in the stromal cells in physiological context and estrogen activity in pathological context such as endometriosis utilizing both genetic and in-vitro evidences, while these evidence remains very descriptive on the phenotype at the absence of GREB1 without deeper mechanism study about how GREB1 interacts with ER and PR to regulate downstream genes in female reproductive tract. Except that, there are also some misinterpretations of the data.

The major concerns are listed below:

- 1 The biggest drawback of current study is the insufficient investigation of the mechanism of GREB1 in uterus. There are already some reports on the mechanism of how GREB1 participate ER and PR function in other systems (PMID: 33731348), the unique role of GREB1 in uterus is rare touched in current study.
- 2 The localization of GREB1 protein detected by IF and IHC is paradoxical. Based on the authors' data,

GREB1 is mainly localized in the nuclear with a puncta pattern, but according to the data from the other paper using Flag-Tag to define the sub-cellular localization (PMID: 33731348), the GREB1 was mainly localized in the cytosol, and in figure 4d, it also seems that GREB1 has both the cytosol and nuclear signal. The author must provide convincing data on GREB1 expression using authenticated antibody.

3 For the mechanism study, though commercially available GREB1 antibodies not work very well, it is feasible to knock-in a tag labeled GREB1 in HESC utilizing the Cas9 or other methods to investigate GREB1 target genes by conducting ChIP-Seq, which is crucial for the validation of proposed mechanism as a cofactor in binding with the target genes.

4 In the whole manuscript, several statements are not very popular in the implantation field. Please check these statements carefully. 1) In line 183-184, the epithelial cell in vitro can respond to E2 as the PR expression in the epithelia was induced by E2 at least in the organoid model. 2) In line 189-190, usually, the progesterone rise starts from D3, and without the implantation, progesterone only will not induce the decidualization. 3) In line 196, the decidual giant cells is confuse, whether the authors referred the polyploid decidual cells? 4) The nidatory estrogen usually referred the D4 morning estrogen just before the implantation, so for Figure 6 and the related description, the authors are misusing this term.

5 In line 236 for the description of MUC1 staining, it seems that the MUC1 was more intensively stained in the KO uteri, the authors should provide more data, such as q-PCR or immunoblot to confirm this.

6 In Figure 5h, indeed the Foxo1 and LIF are not progesterone response genes. Foxo1 was mainly expressed in the epithelial cells in the D4 evening (PMID: 31120913, 30452456) and the expression of LIF was mainly induced by the D4 estrogen (nidatory estrogen). Other more representative progesterone response genes, such as IHH, Areg, HDC in epithelia and Hand2, Hoxa10 in the stromal cell should be considered.

7 In Figure 8c, the epithelial cell from the endometriosis disease was used, whether the epithelial cell isolated from the normal endometrium will also display the similar response to GREB1 knockdown. It should be clearly described in the methods that the stromal cells were isolated in which stage and the culture conditions, since they can respond to estrogen to increase the proliferation

Minor concerns:

1 In Figure 2f, the IP label should be added to make the data more readable.

2 For data collection of ovaries, the stage of estrus should be provided in the methods and figure legend.

3 In line 443, "time-coarse" should be time-course

Response summary to reviewers' comments:

We thank both the Referees for underscoring the importance of our findings as well as the constructive critiques. The constructive feedback provided by both reviewers has been invaluable expanding the depth of our study to strengthen our manuscript. Particularly, we highlight the following new data, which we have included in the revised version:

- In response to Reviewer 1 suggestion, we added the breeding trial analysis for the number of litters between control or *Greb1* KO mice (**Revised Supplementary Fig. S2a**). We also performed the ELISA analysis for serum E2 levels between control and *Greb1* KO mice and found no significant difference between the two groups (**Revised Supplementary Fig. S6**).
- In response to Reviewer 2 concern about the localization of Greb1, we conducted cellular fractionations and immunofluorescence analysis to confirm the nuclear localization of GREB1 in primary HESC cells.
- Overcoming the substantial challenges, we were fortunately able to identify a distinct GREB1 cistrome in HESCs using CUT & RUN sequencing. Additionally, we compared GREB1 peaks with the PR peaks generated by Dr. Demayo's group from similar Cut&Run sequencing in HESCs. From this comparison, we found ~50% of GREB1 bound regions and 63% of GREB1 bound genes also bound by PR, indicating a cofactor role of GREB1 to that of PR (**Revised Fig. 2**).
- As suggested by both the reviewers, we have now measured transcript levels of *Muc1* in uteri *Greb1* KO mice and additional P4-responsive targets, *Hand2*, *Ihh*, *Il13ra2*, and *Cyp26a1* (**Revised Fig. 5 and Supplementary Fig. S3**).
- In addition, we made numerous other changes throughout the text (highlighted with yellow) and figures, as detailed below in our point-by-point response to the reviewers' comments (in blue).

REVIEWER COMMENTS

Reviewer #1 (Remarks to the Author):

Gaining a better understanding of the underlying mechanisms coordinating progesterone and estrogen receptor signaling in the reproductive tract is vital to improving women's reproductive health. The manuscript by Chadchan et al. utilizes a combination of in vitro and in vivo models to investigate the role of GREB1 in regulating endometrial steroid hormone signaling in normal and pathological conditions. The authors demonstrate interesting alterations in GREB1 function between physiological and pathological conditions. Specifically, the manuscript indicates that the deletion of GREB1 results in subfertility stemming from a decreased progesterone response. In comparison, GREB1 is required for estrogen-driven proliferation in a model of endometriosis. Overall, this is a scientifically sound manuscript that will interest reproductive biologists and clinical scientists. The few points I have are detailed below.

1. Line 183. A published report suggests that human endometrial epithelial organoids are responsive to estrogen (PMID: 31666317). The authors may consider the RNA-seq data from that publication.

AUTHORS' RESPONSE: We thank the reviewer for this suggestion. We utilized RNA-seq data from PMID: 31666317 to select additional hormone-responsive genes, as recommended by both reviewers. Furthermore, we have removed the statement from line number 183.

2. Line 189-190. Progesterone does not induce decidualization in the mouse. The sentence needs to be rephrased for accuracy.

AUTHORS' RESPONSE: We thank the reviewer for this suggestion. We rephrased the sentence in the revised manuscript (Line number: 201-205) and modified **Fig. 3** with the correct hormone representation.

3. Line 121- Replace infertility with subfertility.

AUTHORS' RESPONSE: We replaced infertility with subfertility (Line number: 129).

4. The subfertility phenotype warrants further descriptive data from the 6-month breeding trial. (number of litters per animal and time between litters)

AUTHORS' RESPONSE: We now included additional analysis on subfertility phenotype including the number of litters per animal (**Revised Supplementary Fig. S2a**). We also observed irregularities in the time between litters in *Greb1* KO mice (Line number: 225-227).

5. The authors present data indicating no difference in the number of blastocysts recovered from WT and GREB1 KO mice (fig 4), but there is a significant difference in implantation site number (figure 5). In that regard, it is unclear if the embryo orientation defect found on GD 5 (Figure) is observed in the implantation sites demarcated by Chicago blue dye or in all embryo sections.

AUTHORS' RESPONSE: We thank the reviewer for this insightful suggestion. We carefully analyzed all the embryo sections of D5 implantation sites from *Greb1* KO females which revealed a significant embryo orientation defect (Line number: 246-247). These findings imply pleiotropic roles for *Greb1* during peri-implantation and emphasize the need for further investigations.

6. Line 247- *Lif* is an E2-responsive gene on gestational day 4.

AUTHORS' RESPONSE: We thank the reviewer for noting this. We now removed this gene from the P4-target genes panel.

7. In the endometriosis model, it is essential that the authors validate that there is no difference in circulating E2 levels between global GREB1 KO and WT mice. The reviewer understands that serum may not have been saved from this experiment and believes it would be acceptable to determine circulating estrogen levels in a separate cohort of intact mice at estrus.

AUTHORS' RESPONSE: We appreciate the reviewer for bringing this to our attention. We conducted an ELISA and observed no significant difference in serum E2 levels between *Greb1* KO and WT endometriotic mice (**Revised Supplementary Fig. S6**).

Reviewer #2 (Remarks to the Author):

The cell type specific responses to the steroid hormones and the intricate interactions between ER and PR still remain elusive in female reproduction with little progress have made in recent decade. The manuscript entitled "A distinct feed-forward mechanism between GREB1 and steroid receptors dictates hormone dependent GREB1 action in endometrial function and dysfunction" describe the pathophysiologic function of GREB1 in female endometrium. The authors proposed that GREB1 mainly mediate progesterone activity in the stromal cells in physiological context and estrogen activity in pathological context such as endometriosis utilizing both genetic and in-vitro

evidences, while these evidence remains very descriptive on the phenotype at the absence of GREB1 without deeper mechanism study about how GREB1 interacts with ER and PR to regulate downstream genes in female reproductive tract. Except that, there are also some misinterpretations of the data. The major concerns are listed below:

1 The biggest drawback of current study is the insufficient investigation of the mechanism of GREB1 in uterus. There are already some reports on the mechanism of how GREB1 participate ER and PR function in other systems (PMID: 33731348), but the unique role of GREB1 in uterus is rare touched in current study.

AUTHORS' RESPONSE: We appreciate the reviewer's constructive critique. While the PMID: 33731348 report made a significant discovery regarding the ER α -GREB1 nexus in mammary tumors, it did not delve into the GREB1 and PR connection. Notably, much of GREB1's attributed function is linked to ER actions in MCF-7 cells, lacking insights on the cell type-specific functions of GREB1 in any physiologies. Our study unveiled the distinct roles of GREB1 in uterine physiological responses, revealing several intricate cellular mechanisms. First, GREB1 null mice displayed a normal HPA axis, intact ovarian function, and typical embryo development. Second, we attributed GREB1's action specifically to uterine receptivity and confined to stromal-derived cells. Second, we discovered that GREB1 does not regulate E2 action but rather mediates P4 action for controlling uterine receptivity. Third, we demonstrated GREB1's interaction with PR and its regulatory role in PR binding on chromatin. Further, in the revised version, we identified GREB1 cistrome and co-occupancy with PR on specific genomic regions, revealing its cofactor role in uterine physiology.

In terms of pathological significance, despite reported GREB1 SNPs in endometriosis, their functional relevance remained unexplored until our study. We respectfully argue that our study introduces a novel nexus where a downstream mediator of E2/P4 functions differentially in physiological or pathological contexts within the same tissue through the same feed-forward axis.

It is important to note that exploring into the intricate mechanisms by which GREB1/PR/ER coordinate the cell-type specific actions of E2 or P4 in the uterus necessitates a considerable duration of research. For example, generating murine models by deleting the PRE or EREs alone or in combination on murine Greb1 locus will unearth the complex nexus between GREB1/ER/PR. We respectfully acknowledge that studies of this nature are currently beyond the scope of this manuscript, particularly considering the 4-5 years it took to generate GREB1 KO mice during my tenure in Dr. Bert O'Malley's group.

2 The localization of GREB1 protein detected by IF and IHC is paradoxical. Based on the authors' data, GREB1 is mainly localized in the nuclear with a puncta pattern, but according to the data from the other paper using Flag-Tag to define the sub-cellular localization (PMID: 33731348), the GREB1 was mainly localize in the cytosol, and in figure 4d, it also seems that GREB1 has both the cytosol and nuclear signal. The author must provide convincing data on GREB1 expression using authenticated antibody.

AUTHORS' RESPONSE: We acknowledge the reviewer's concern, but it is noteworthy that, with the exception of PMID: 33731348, all other studies, including PMIDs: 23403292; 31462641; and 30644358, consistently reported the nuclear localization of GREB1. Specifically, PMID: 23403292 highlighted the exclusive nuclear localization of Flag-GREB1 as puncta, with the deletion of the NLS signal resulting in GREB1 shuttling into the cytoplasm. Importantly, the Flag-GREB1 construct in PMID: 33731348 cloned from MCF-7 cells cDNA, raising the possibility of cell type-specific and/or system-specific localization variations. Nonetheless, to address this concern, we conducted additional experiments involving nuclear and cytoplasmic protein extractions from primary HES cells, coupled with immunofluorescence analysis using GREB1 siRNA to validate the subcellular localization. Our results consistently reveal nuclear accumulation of GREB1 following MPA treatment (**Fig 1**). While modest expression of GREB1 in the cytoplasmic fraction observed in control cells, we could not readily detect the same with MPA treatment. Furthermore, IF analysis confirmed GREB1 localization to nuclear puncta, and this pattern was completely abolished with GREB1 siRNA (**Fig 2**). These data support that, in primary HESC cells, GREB1 predominantly localizes to the nucleus and validates tested antibody.

Fig 1: Western blotting to show the GREB1 expression in nuclear or cytoplasmic endometrial stromal cells lysate.

Fig 2: Immunofluorescence to show GREB1 expression in MPA-treated control or GREB1 siRNA transfected endometrial stromal cells.

3 For the mechanism study, though commercially available GREB1 antibodies not work very well, it is feasible to knock-in a tag labeled GREB1 in HESC utilizing the Cas9 or other methods to investigate GREB1 target genes by conducting ChIP-Seq, which is crucial for the validation of proposed mechanism as a cofactor in binding with the target genes.

AUTHORS' RESPONSE: We agree with the reviewer regarding the significance of GREB1 binding peaks on target genes. Acknowledging the importance of this aspect and despite multiple unsuccessful attempts with Active Motif, Washington University, and Baylor GARP core, we went for both the ChIP-Seq and CUT&RUN methods for GREB1. However, we encountered several challenges in creating tagged-GREB1 HESC. Our efforts to obtain published FLAG-GREB1 constructs from two investigators were unsuccessful despite repeated requests. Fortunately, the advanced CUT&RUN method proved effective for the GREB1 protein in identifying a distinct GREB1 cistrome in HESCs (**Revised Fig. 2**). Additionally, we found that 50% of GREB1 peaks are bound by PR and 63% of GREB1 bound genes are also bound by PR (**Revised Fig. 2**). These results clearly indicate that GREB1 is a cofactor of PR in the endometrium, supporting the overall findings of this manuscript.

4 In the whole manuscript, several statements are not very popular in the implantation field. Please check these statements carefully. 1) In line 183-184, the epithelial cell in vitro can response to E2 as the PR expression in the epithelia was induced by E2 at least in the organoid model. 2) In line 189-190, usually, the progesterone rise starts from D3, and without the implantation, progesterone only will not induce the decidualization. 3) In line 196, the decidual giant cells is confuse, whether the authors referred the polyploid decidual cells? 4) The nidatory estrogen usually referred the D4 morning estrogen just before the implantation, so for Figure 6 and the related description, the authors are misusing this term.

AUTHORS' RESPONSE: We thank the reviewer for suggestions on these statements. We have now rephrased the aforementioned sentences as detailed below:

1) Line 183-184- We agree with the reviewer that some recent studies provided evidence that endometrial epithelial organoids are responsive to E2 or P4. We removed that statement from line number 183.

2) Line 189-190 – We rephrased the text (New line number:201-205).

3) In line 196- We corrected the typo.

4) In Figure 6, we replaced the nidatory estrogen with estrogen only.

5 In line 236 for the description of MUC1 staining, it seems that the MUC1 was more intensively stained in the KO uteri, the authors should provide more data, such as q-PCR or immunoblot to confirm this-

AUTHORS' RESPONSE: We confirmed the *Muc1* levels by q-PCR in uteri from pregnant mice and found no significant difference in expression between control and *Greb1* KO mice (**Revised Fig. S3a**).

6 In Figure 5h, indeed the *Foxo1* and LIF are not progesterone response genes. *Foxo1* was mainly expressed in the epithelial cells in the D4 evening (PMID: 31120913, 30452456) and the expression of *Lif* was mainly induced by the D4 estrogen (nidatory estrogen). Other more representative progesterone response genes, such as *IHH*, *Areg*, *HDC* in epithelia and *Hand2*, *Hoxa10* in the stromal cell should be considered.

AUTHORS' RESPONSE: We appreciate the reviewer for noting this and the suggested targets. In Figure 5h, we have incorporated additional P4-regulated targets, such as *Ihh*, *Il13ra2*, and *Cyp26A1*, and found that these targets were significantly downregulated in *Greb1* KO uteri when compared to WT controls (**Revised Fig. 5 and Supplementary Fig. S3**). We also would like to note that multiple studies identified *FOXO1* as a progesterone-inducible gene in the endometrium, as reported in these publications (PMIDs: 23381604, 31665330, 18096667).

7 In Figure 8c, the epithelial cell from the endometriosis disease was used, whether the epithelial cell isolated from the normal endometrium will also display the similar response to GREB1 knockdown. It should be clearly described in the methods that the stromal cells were isolated in which stage and the culture conditions, since they can response to estrogen to increase the proliferation

AUTHORS' RESPONSE: We appreciate the reviewer for bringing up this point. We hypothesize that epithelial cells from the normal endometrium may not exhibit a similar response to GREB1 knockdown, as evidenced by the lack of a significant difference in proliferation observed when control or *Greb1* KO mice were treated with E2 (**Fig. 6b**). The stromal cells utilized in Figure 8c are primary cells isolated from endometriotic lesions that were collected from consenting women undergoing surgery at the clinic. Unfortunately, clinician collaborators did not provide the stage of the cycle details. As suggested, we have now included the culture conditions for stromal cell isolation in the Methods section of the revised version (Line numbers: 451-460).

Minor concerns:

1 In Figure 2f, the IP label should be added to make the data more readable.

AUTHORS' RESPONSE: We added an IP label to make it more readable (**Revised Fig. 2e**).

2 For data collection of ovaries, the stage of estrus should be provided in the methods and figure legend.

AUTHORS' RESPONSE: We now provided the stage of estrus in methods and figure legend as suggested in the revised manuscript (Line number: 582-589; 30-31 Supplementary material file).

3 In line 443, "time-coarse" should be time-course

AUTHORS' RESPONSE: We corrected the typo error.

REVIEWERS' COMMENTS

Reviewer #1 (Remarks to the Author):

The authors have adequately addressed my comments, and the manuscript has improved considerably since the initial submission.

Reviewer #2 (Remarks to the Author):

The author performed substantial work to address my concerns. There are still some concerns remained.

1 For Fig 1F, Color bar show be provided.

2 For Fig 1F, it is better to calculate the enrichment of GREB1 around TSS, but not the start site to end site.

3 Still, the localization of GREB1 in Fig 3B (IF) and Fig 4D (IHC) in mouse uteri is very contradictory. IHC result shows that there is robust expression of GREB1 in cytoplasm, except its nuclear expression.

REVIEWER COMMENTS

Reviewer #1 (Remarks to the Author):

The authors have adequately addressed my comments, and the manuscript has improved considerably since the initial submission.

AUTHORS' RESPONSE: We express our gratitude to the reviewer for the positive response on our manuscript. Your constructive feedback and guidance throughout the review process have been invaluable, contributing to the enhancement of our work.

Reviewer #2 (Remarks to the Author):

The author performed substantial work to address my concerns. There are still some concerns remained.

1 For Fig 1F, Color bar show be provided.

AUTHORS' RESPONSE: We presume the reviewer meant Fig 2F as there is no 1F. We have now added the color bar in Fig. 2F.

2 For Fig 1F, it is better to calculate the enrichment of GREB1 around TSS, but not the start site to end site.

AUTHORS' RESPONSE: The number of ChIP-Seq/CUT&RUN studies for nuclear receptors indicates that many of these receptors bind to distal intergenic regions, introns, and enhancers. Similarly, GREB1 binds to intergenic regions, introns, and promoters. Therefore, we chose to display all GREB1 peaks in Fig. 2F.

3 Still, the localization of GREB1 in Fig 3B (IF) and Fig 4D (IHC) in mouse uteri is very contradictory. IHC result shows that there is robust expression of GREB1 in cytoplasm, except its nuclear expression.

AUTHORS' RESPONSE: We respectfully disagree with the reviewer regarding the robust expression of GREB1 in the cytoplasm. Multiple pieces of evidence from our cell fractionation assays, along with findings from previous studies (PMIDs: 23403292; 31462641; and 30644358), clearly demonstrate the predominant expression of GREB1 in the nucleus rather than the cytoplasm. While we acknowledge the presence of a slight staining in the cytoplasm in Fig. 4D IHC, it is crucial to consider that in GREB1 KO mice uteri (middle panel), there is also a slight cytoplasmic staining, suggesting that this could be background staining. It is important to note that in Fig 3B IF, there is a clear and distinct nuclear staining observed in uteri from pregnant mice, indicating that GREB1 localizes to the nucleus in the endometrium.